# Functional convergence of genomic and transcriptomic architecture underlies schooling behaviour in a live-bearing fish

Alberto Corral-Lopez [1,2,3] ✉, Natasha I. Bloch[4], Wouter van der Bijl [1], Maria Cortazar-Chinarro[5,6], Alexander Szorkovszky[7], Alexander Kotrschal [8], Iulia Darolti[1,9], Severine D. Buechel[8], Maksym Romenskyy[2], Niclas Kolm [2] & Judith E. Mank [1]

The organization and coordination of fish schools provide a valuable model to investigate the genetic architecture of affiliative behaviours and dissect the mechanisms underlying social behaviours and personalities. Here we used replicate guppy selection lines that vary in schooling propensity and combine quantitative genetics with genomic and transcriptomic analyses to investigate the genetic basis of sociability phenotypes. We show that consistent with findings in collective motion patterns, experimental evolution of schooling propensity increased the sociability of female, but not male, guppies when swimming with unfamiliar conspecifics. This finding highlights a relevant link between coordinated motion and sociability for species forming fission–fusion societies in which both group size and the type of social interactions are dynamic across space and time. We further show that alignment and attraction, the two major traits forming the sociability personality axis in this species, showed heritability estimates at the upper end of the range previously described for social behaviours, with important variation across sexes. The results from both Pool-seq and RNA-seq data indicated that genes involved in neuron migration and synaptic function were instrumental in the evolution of sociability, highlighting a crucial role of glutamatergic synaptic function and calcium-dependent signalling processes in the evolution of schooling.

Living in groups, a widespread phenomenon across the animal kingdom, can lead to strikingly complex social behaviours, such as cooperative interactions, subdivision of labour or collective decision-making[1]. Sociability, the propensity to affiliate with other animals, can also vary across individuals. Sociability represents a fundamental aspect of personality which can influence social interactions and is often subject to strong selective processes[2,3]. Indeed, intraspecific differences in sociability are widespread (for example, refs. [4,5]) and individual genetic

[1]Department of Zoology and Biodiversity Research Centre, University of British Columbia, Vancouver, British Columbia, Canada. [2]Department of Zoology/Ethology, Stockholm University, Stockholm, Sweden. [3]Division of Ecology and Genetics, Uppsala University, Uppsala, Sweden. [4]Department of Biomedical Engineering, University of Los Andes, Bogota, Colombia. [5]Department of Earth, Ocean and Atmospheric Sciences, University of British Columbia, Vancouver, British Columbia, Canada. [6]MEMEG Department of Biology, Lund University, Lund, Sweden. [7]RITMO Centre for Interdisciplinary Studies in Rhythm, Time and Motion, University of Oslo, Oslo, Norway. [8]Behavioural Ecology, Wageningen University and Research, Wageningen, the Netherlands. [9]Department of Ecology and Evolution, University of Lausanne, Lausanne, Switzerland. ✉e-mail: alberto.corral@ebc.uu.se

variation often underlies variability in personality and social behaviour phenotypes[6]. However, heritability estimates of social behaviour traits are consistent with a complex, polygenic architecture[7]. Human twin and family studies reveal that heritability estimates of personality traits generally ≈0.40 (reviewed in ref. [8]). In non-human animals, a meta-analysis estimated that mean heritability is 0.23 across social behaviours, including personality traits[9], with heritability of affiliative associations ranging substantially from 0.11 to 0.51 (refs [10]–[13]).

Despite this complexity, multiple neural and genetic mechanisms underlying social behaviour have been identified[14]. Many of the neural structures and neuromodulators (serotonin, dopamine, vasopressin and oxytocin) are highly conserved within the social decision-making network across vertebrates[15]. Moreover, human personality traits associated with social decision-making have been linked to dopaminergic and serotonergic genes (reviewed in ref. [16]), and the regulation of these neuromodulators has been connected to neurodevelopmental disorders that affect affiliative behaviours, such as autism spectrum disorder[6,17,18]. Studies in non-human organisms likewise point towards a major role of genes involved in the regulation of these neurochemical systems. For instance, mouse knockout mutants for genes involved in dopaminergic signalling exhibit altered sociability phenotypes[19], and changes in sociability in three-spined sticklebacks, *Gasterosteus aculeatus*, are predicted by natural variation in the expression of genes within the dopaminergic and stress pathways[20]. However, while specific groups of genes have been identified for a range of affiliative behaviours, we lack a deeper understanding of their role in inter-individual variation and evolutionary processes underlying sociability.

In fish, group living often leads to spectacular forms of collective behaviour, with members of a school coordinating their movements to increase efficiency in foraging, travelling or predator avoidance[1]. The extent to which members of a school coordinate their movements is an integral part of the sociability axis of personality, that is, how individuals react to the presence or absence of conspecifics excluding aggressive behaviours[21]. We previously showed that schooling behaviour has a repeatability of 0.43 at the individual level[22] and this can increase substantially over just three generations of artificial selection in female guppies, *Poecilia reticulata*, generating a 15% increase in intrinsic schooling propensity compared with controls[22,23]. Selection was based on a group phenotype, polarization, or the level of alignment between individuals moving together in a group.

Understanding the genetic basis of this schooling phenotype requires linking individual phenotypic differences to genetic variation. In this study, we phenotyped alignment and attraction of 1,496 guppies across 195 families (father, mother, three female and three male offspring from our replicate experimental selection lines) to estimate the heritability of these two motion characteristics that previous factor analyses identified to be integral components for the sociability axis of personality in this species[21]. Because many social interaction patterns in guppies have sex differences[24–26], and because our selection was performed solely on females, we are able to examine cross-sex genetic correlations in this ecologically relevant behavioural trait. Genomic and transcriptomic data from these lines reveal convergence in the genetic architecture of sociability, highlighting a series of genes with well-defined roles in neurodevelopmental processes. Our results provide a robust agreement across experiments about the genetic regulation of neural processes in decision making and motor control regions of the brain, and its importance for variation of personality within individuals of this species.

## Results

### Heritability of sociability in guppies

We first determined whether experimental evolution for higher schooling propensity affected social interactions with unfamiliar conspecifics. For this, we assessed sociability in 740 females and 746 males from multiple families of three replicate lines artificially selected for a 15%

average higher polarization (polarization-selected lines hereafter) and three replicate control lines exposed to a group of non-kin unfamiliar conspecifics in an open field test. Specifically, we quantified their alignment and nearest neighbour distance (attraction), two measures of collective motion characteristics that are demonstrated to capture the most biologically relevant aspects of the sociability axis of personality in this species[21].

Female guppies from polarization-selected lines presented higher alignment and higher attraction to an unfamiliar group compared with control lines (linear mixed model for alignment, $\text{LMM}_{\text{alignment}}$: line: $t = 2.27$, d.f. = 9.68, $P = 0.047$; $\text{LMM}_{\text{attraction}}$: line: $t = -2.34$, d.f. = 9.41, $P = 0.043$; Fig. 1a and Supplementary Table 1). No differences were observed in these traits between polarization-selected and control males ($\text{LMM}_{\text{alignment}}$: line: $t = -1.38$, d.f. = 9.56, $P = 0.20$; $\text{LMM}_{\text{attraction}}$: line: $t = 0.88$,; d.f. = 9.26, $P = 0.40$; Fig. 1a and Supplementary Table 2). Our analyses showed an effect of sex in alignment, with females exhibiting ~8% higher alignment than males ($\text{LMM}_{\text{alignment}}$: sex: $t = -3.02$, d.f. = 690.08, $P = 0.003$), but no difference between sexes in attraction to a group of unfamiliar conspecifics ($\text{LMM}_{\text{attraction}}$: sex: $t = 0.51$, d.f. = 447.05, $P = 0.61$; Fig. 1a and Supplementary Table 1). There were some differences in sociability between the parental and offspring generation tested in our experiment, with higher alignment to group average direction and lower distances to nearest neighbour observed in offspring ($\text{LMM}_{\text{alignment}}$: generation: $t = -10.13$, d.f. = 1141.24, $P < 0.001$; $\text{LMM}_{\text{attraction}}$: generation: $t = 11.29$, d.f. = 992.16, $P < 0.001$; Fig. 1a and Supplementary Table 1). Differences in body size between age classes in guppies may explain these results (see Supplementary Table 3), as the time restrictions involved in testing large numbers of fish required that we assessed individuals from the parental and offspring generations at different ages. However, these differences are unlikely to create large biases in our heritability estimates given that we tested all fish after sexual maturation and that polarization-selected and control fish were of similar age within parents tested (9 months old) and within offspring tested (5 months old). In addition, the difference in means between generations is accounted for in our statistical models (see Methods).

To assess the heritability of sociability in this species, we fitted animal models with alignment and attraction phenotypes quantified from these 1,486 individuals comprising parents, three male and three female offspring for 195 families (99 polarization-selected and 96 control families). Given known differences between the sexes in social interaction patterns in guppies[24–26], we estimated heritability with animal models that only included relationships with same-sex individuals (same-sex pedigree) or that included relationships with individuals from both sexes (full pedigree).

Using same-sex pedigree animal models, attraction heritability was similar in females ($h^2_{\text{attraction}}$, estimate (95% credible interval (CI)) = 0.18 (0.05, 0.34); Fig. 1b) and males ($h^2_{\text{attraction}} = 0.19$ (0.06, 0.34); Fig. 1b); however, alignment heritability was much higher in females ($h^2_{\text{alignment}} = 0.34$ (0.18, 0.49); Fig. 1b) than in males ($h^2_{\text{alignment}} = 0.06$ (0.00, 0.18); Fig. 1b). Full-pedigree models indicated lower heritability estimates than same-sex pedigree models (Supplementary Table 5 and Fig. 1b), except for the heritability estimate of attraction in males ($h^2_{\text{attraction}} = 0.26$ (0.16, 0.37); Fig. 1b). Finally, animal models indicated a positive female–male genetic correlation in attraction ($r_{\text{f–m, attraction}}$: 0.68 (0.23, 0.98); Fig. 1b), although the magnitude of this correlation contained large CIs. For alignment, CIs for $r_{\text{f–m}}$ are also wide and span zero ($r_{\text{f–m, alignment}}$: 0.44 (−0.17, 0.95)), and we can only conclude that the cross-sex genetic correlation is not strongly negative (Supplementary Table 5 and Fig. 1b).

### Genetic basis of sociability in guppies

Our quantitative genetic analyses of alignment and attraction suggest an important genetic influence on sociability phenotypes of guppies. As such, we sequenced DNA pools (Pool-seq) to identify genome-wide differences in allele frequencies between polarization-selected female

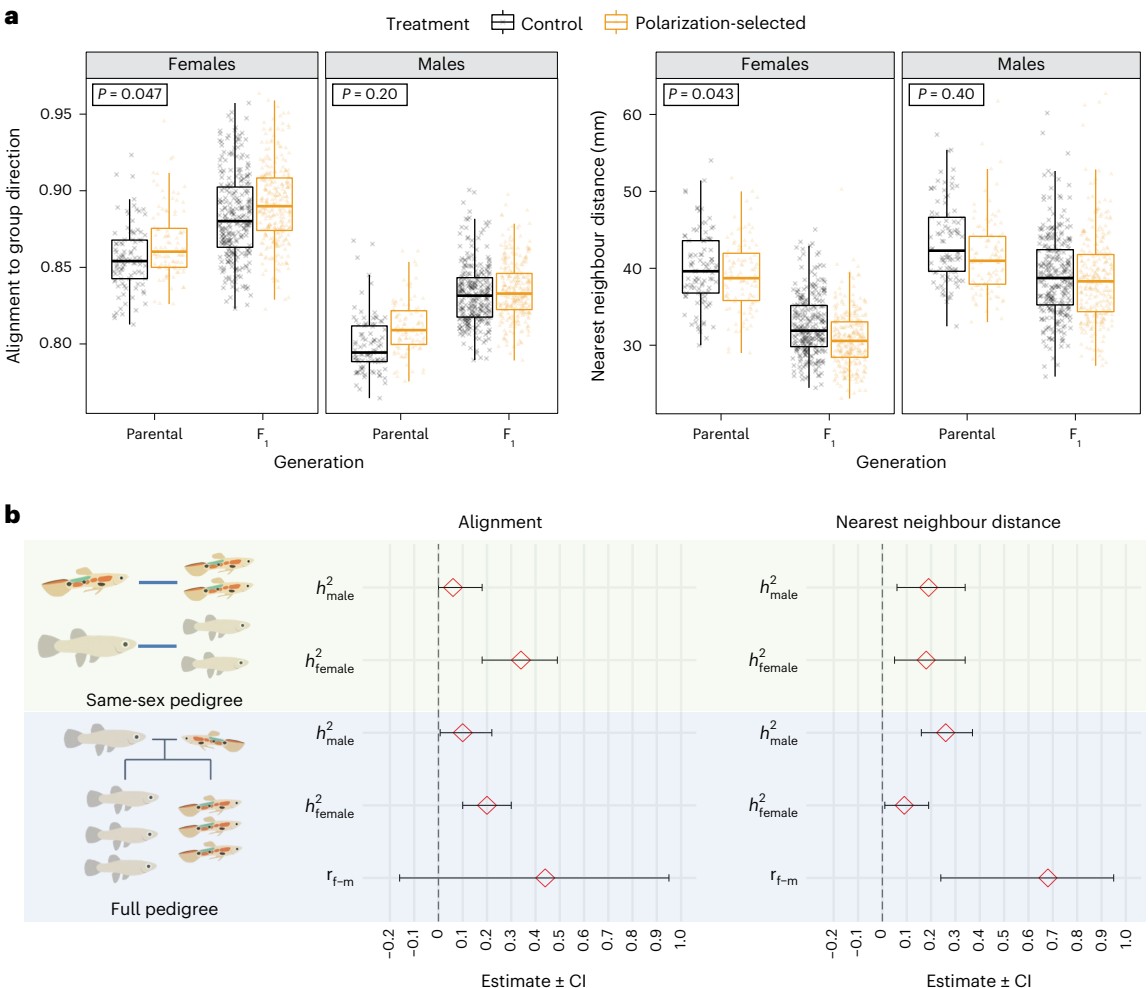

**Fig. 1 | Heritability of sociability in guppies. a**, Female, but not male, guppies from polarization-selected lines (*n* = 763, orange) presented higher alignment to the group direction (left) and shorter distances to their nearest neighbour (higher alignment; right) than guppies from control lines (*n* = 724, grey) when swimming with unfamiliar same-sex conspecifics (see Supplementary Tables 1 and 2). For all boxplots, horizontal lines indicate medians, boxes indicate the interquartile range and whiskers indicate all points within 1.5 times the interquartile range. Boxes in top left position of each facet indicate Tukey adjusted *P* values for multiple contrasts (*P* < 0.05 in bold) for statistical contrasts by sex in an LMM comparing alignment and attraction between selection line treatments (see Supplementary Tables 1 and 2). **b**, Animal models using same-sex pedigrees and full pedigrees with alignment and attraction (nearest neighbour distance) phenotypes in 195 families of polarization-selected and control guppy lines indicated a moderate heritability in female guppies for both biologically relevant aspects of sociability measured, alignment (left) and attraction (right). In males, we found moderate heritability in attraction, but CIs in alignment estimates overlapped with 0, suggesting low heritability of this sociability aspect. Our full-pedigree animal models provided large CIs for male−female correlations in sociability, with estimates overlapping 0 in alignment, but a positive correlation between sexes in attraction (see Supplementary Tables 4 and 5). Red diamonds indicate mean heritability values with 95% CIs.

guppies that presented high sociability and polarization-selected females that presented low sociability. Specifically, we focused on measurements obtained from females in analyses of alignment to an unfamiliar group. For this, we pooled the DNA from mothers whose families (normalized mother and daughters alignment score; see Methods) were in the top 25% and the bottom 25% quartiles from each of the three replicated polarization-selected lines (six total pooled samples with 7 individuals each; Supplementary Fig. 1).

DNA reads were aligned to the guppy reference genome (Guppy_female_1.0 + MT, RefSeq accession: GCA_000633615.2) to compare genome-wide allele frequency differences between high- and low-sociability guppies. We ran two independent analyses with these aligned sequences. For our first analysis, we merged sequences from the three replicates with high-sociability pooled samples and sequences from the three replicates with low-sociability pooled samples. We filtered merged sequences to 3,004,974 single nucleotide polymorphisms (SNPs; see Methods) and performed a Fisher's exact

test in Popoolation2 (ref. 27) to identify SNPs that significantly differed in their allele frequencies between guppies with high and low sociability. Using this methodology, we identified 819 SNPs associated with our sociability phenotype (Fisher's exact test, *P* < 10$^{-8}$; Fig. 2a). SNPs over this standard genome-wide significance threshold[28] were mostly found in single physically unlinked positions across the genome, consistent with a polygenic architecture of the trait.

Out of these 819 significantly different SNPs, 421 were located within genes or gene promoter regions of the guppy genome and were used for further functional characterization in association with Gene Ontology (GO) annotations (273 unique genes). We clustered GO terms on the basis of semantic similarities and found significant overrepresentation of biological process terms related to learning and memory, synaptic functioning, response to stimulus, locomotion and growth (Fig. 2b). We likewise found significant overrepresentation of cadherin and calcium-dependent protein binding annotations (molecular components terms; Supplementary Fig. 2)

and glutamatergic synapse annotations (cellular components terms; Supplementary Fig. 3).

Second, we looked for consistent differences in allele frequencies between high- and low-sociability pooled samples in our three replicates by performing the Cochran–Mantel–Haenszel test (CMH test) in Popoolation2 (ref. 27). Convergent changes in allele frequency probably represent selected sites and are less likely the result of genetic drift in any one line. This stringent analysis identified 13 SNPs from 10 different chromosomes with consistent significant differences in allele frequencies across the three replicates (CMH test $P < 0.01$ with false discovery rate (FDR) correction). Five of these SNPs are located within known coding sequence of the guppy genome, of which three are within well-characterized genes in zebrafish and human homologues with important roles for cognitive function: ubiquitin-specific peptidase 11 (*usp11*), *supt6* histone chaperone and transcription elongation factor homologue (*supt6h*) and cadherin 13 (*cdh13*; Fig. 2a and Table 1). The other two are classified as novel genes, one of them being matched to an RNA-binding protein *Nova-1*-like gene, similarly associated with motor function and changes in synaptic function (Fig. 2a and Table 1).

### Neurogenomic response of schooling in guppies

We used transcriptome sequencing to determine differences in gene expression in multiple brain regions of polarization-selected and control females in response to two different social contexts, swimming alone (the 'Alone' condition) or schooling in a group (groups of eight unfamiliar females; the 'Group' condition). We focused on three separate brain tissues that control distinct functions. The 'optic tectum' is involved in sensory processing of visual signals. The 'telencephalon' is implicated in decision making. The 'midbrain' is associated with motor function in response to auditory and visual stimuli[29,30]. Together, these three brain tissues contain the main components of the social brain network in fish[31,32].

**Differential expression analyses.** We identified genes differentially expressed between lines under each treatment condition and in each brain region separately to determine the neurogenomic response triggered by schooling in both lines. Gene expression analyses indicated very little overlap in differentially expressed (DE) genes between polarization-selected and control lines (Fig. 3 and Supplementary Data). Specifically, we found that only adipocyte enhancer-binding protein 2 gene (*AEBP2;* involved in adipocyte differentiation) in the midbrain and an unknown gene in the optic tectum were differentially expressed in both polarization-selected and control lines. Such little overlap suggests that females from different selection lines are activating different transcriptional cascades and biological pathways in response to social context. In polarization-selected lines we found an order of magnitude fewer DE genes in the optic tectum than in the other brain components ($n = 21$ for optic tectum, $n = 158$ for telencephalon, $n = 109$ for midbrain, each $P_{adj} < 0.05$). Moreover, in the telencephalon and midbrain, DE genes between Alone and Group treatment in the polarization-selected lines were enriched for GO annotations associated with cognition, memory, learning and social behaviour (Supplementary Data). We found enrichment for these annotation terms for DE genes expressed in the optic tectum but not in the midbrain or telencephalon of control lines.

Hierarchical clustering analyses of DE genes showed that females from polarization-selected lines in the Group condition clustered uniquely from polarization-selected females in the Alone condition (Fig. 3 and Supplementary Table 6). Similarly, females from polarization-selected lines in the Group condition clustered uniquely from females from control lines under the Group and Alone conditions in both the telencephalon and the midbrain, suggesting a unique response in the regions of the brain associated with behaviour to social

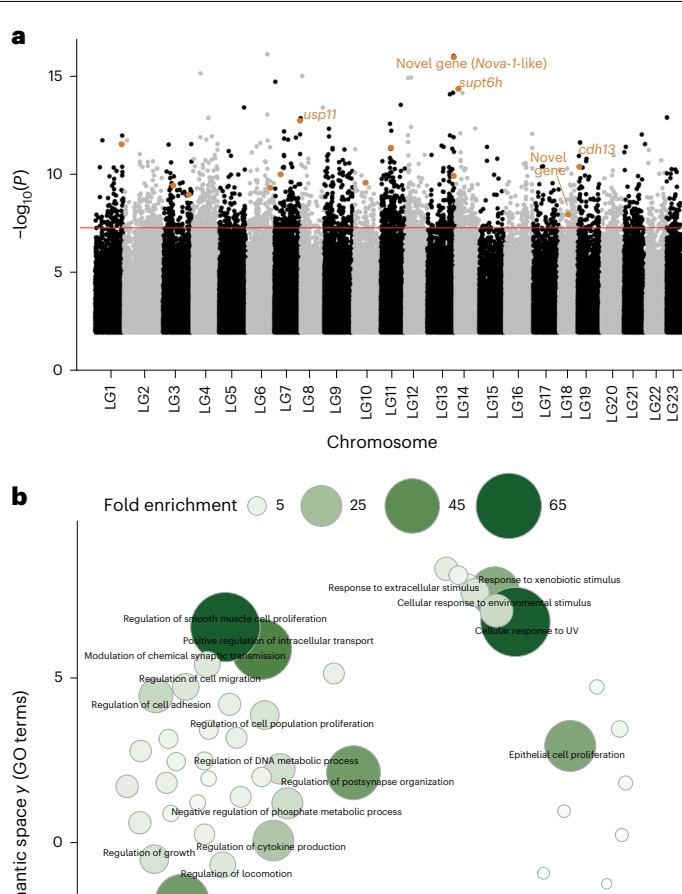

**Fig. 2 | Genetic basis of sociability in the guppy. a**, Manhattan plot of $-\log_{10}(P)$ values across linkage groups (LG) in the guppy genome resulting from a two-sided Fisher's exact test comparing allele frequency differences between high- and low-sociability female guppies. We merged pooled DNA sequences of three independent replicates and found 819 SNPs to be significantly different (above genome-wide threshold highlighted in red), while a highly stringent analyses of consistent allele frequency differences across our three independent replicates (CMH test; see Methods) identified 13 SNPs (5 of them within genes) associated with sociability in the species (gene names and SNP location in the genome highlighted in orange). SNPs with $-\log_{10}(P) < 2$ are omitted. **b**, Clustering of statistically significant overrepresented GO annotations for biological processes associated with differences between high and low sociability in guppies. Point size and colour provide information on fold enrichment value from the statistical overrepresentation test performed in PANTHER[84] (see Methods). Terms with fold enrichment lower than 8 are represented but not described in text. Axes have no intrinsic meaning and are based on multidimensional scaling which clustered terms based on semantic similarities[74].

exposure. This was not observed in samples from the optic tectum, suggesting that visual processing of social treatments did not differ between polarization-selected and control females. Hierarchical clustering analyses using all expressed genes clustered samples by selection line rather than by social context condition (Supplementary Fig. 4), suggesting that social context affects only a targeted subset of the overall transcriptome rather than the majority of genes.

**Table 1 | Characterization of genes associated with sociability in guppies**

| SNP location | Gene ID Ensemble | Gene name | Described cognitive function of homologues | References |
|---|---|---|---|---|
| Chr 7: 30748139 | 00000004543 | *usp11* | Control of cortical neurogenesis and neuronal migration Mutations of the gene have been associated with neurological disorders. | ref. 44 ref. 91 |
| Chr 13: 31383940 | 00000018946 | Novel gene (RNA-binding protein *Nova-1*-like) | Neuronal RNA-binding protein associated with motor function | ref. 46 |
| Chr 14: 4286109 | 00000009725 | *supt6h* | Substrate of mTOR, a signalling pathway associated with brain function and neurodegenerative disorders | ref. 42 ref. 43 |
| Chr 18: 4286109 | 00000014318 | Novel gene | – | – |
| Chr 19: 3032099 | 00000019822 | *cdh13* | Modulation of brain activity through GABAergic function Organization of neuronal circuits | ref. 47 ref. 92 |

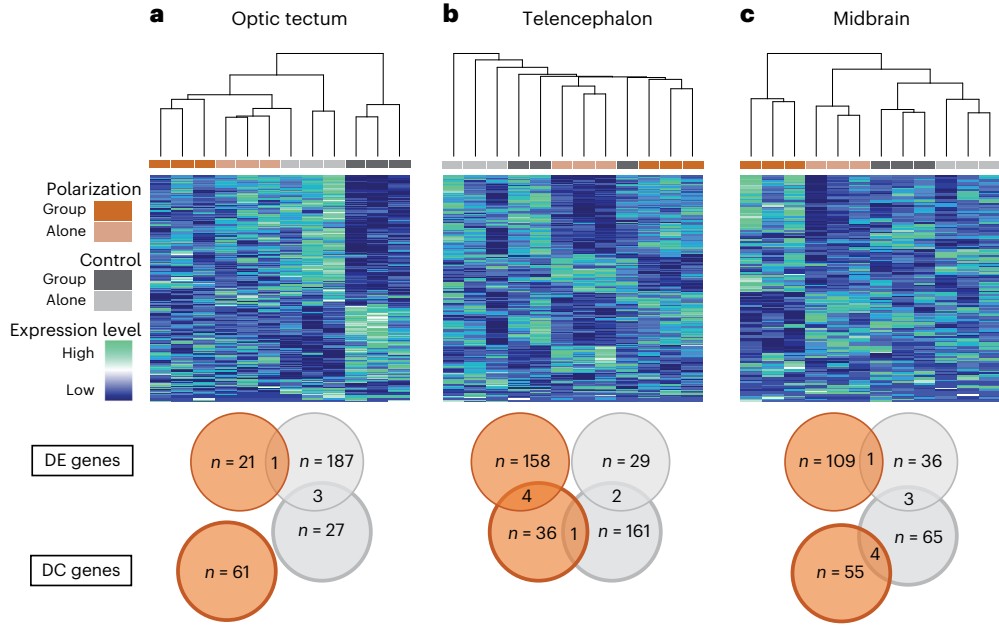

**Fig. 3 | Neurogenomic response of schooling in guppies. a–c**, Hierarchical clustering and relative expression levels for all differentially expressed genes between Alone and Group treatments in the optic tectum (**a**), the telencephalon (**b**) and the midbrain (**c**). Differentially expressed genes were identified separately in polarization and control line samples. Clustering, based on Euclidian distance, represents transcriptional similarity across all samples. Venn diagrams summarize the total number of DE genes and DC gene pairs in each tissue for polarization-selected and control lines.

**Differential co-expression analyses.** We used systems biology methods designed to compare the co-expression networks between conditions to identify genes that change in the way they are connected to other genes within the co-expression network across conditions, independent of whether they are differentially expressed[29–31]. Specifically, we used Bayes approach for differential co-expression analysis (BFDCA)[30] to identify differentially co-expressed (DC) gene pairs under the Group and Alone conditions (that is, pairs of genes that significantly change in correlation between the two social contexts for each line[30,32]). Similar to the findings in the DE analyses, we found little overlap in the genes forming DC gene pairs between comparisons implemented for control and polarization-selected lines (see Supplementary Tables 7 and 8, and Fig. 3). Together, our results suggest that polarization-selected lines were activating different biological pathways compared with control lines to modulate coordinated movement.

We additionally found a group of genes that are both DE and DC in the same tissue and line, suggesting that they might play an important role in mediating coordinated movement (Supplementary Tables 8 and 9). Specifically, in the telencephalon, we identified 4 genes that are both DE and DC in polarization-selected lines: *LRRC24*, *PTPRS*, *KHDR2* and *PP2BA* (Supplementary Table 9). These genes are part of

the calcineurin and the Wnt/oxytocin signalling pathways known to be involved in modulating social behaviour, learning and memory[33–35]. Enrichment tests confirm the functional relevance of the DC gene pairs identified, revealing an overrepresentation of genes associated with the glutamatergic synapse, as well as with visual transduction among DC gene pairs in multiple comparisons (Supplementary Tables 10 and 11).

**Functional characterization of genes of interest across experiments**

We combined the information from our genomic and transcriptomic analyses on polarization-selected and control lines to obtain an intersected delimitation of the gene functions that our analyses highlighted as important in the development and expression of social interactions with conspecifics. Specifically, we used functional analyses in the set of genes with differentiated SNPs between merged sequences of the three replicates with high and low sociability (273 unique genes) as reference and compared results to functional analyses of genes differentially expressed in three different brain tissues of females following exposure to multiple social conditions. We found a concordance of 79% in the combination of biological processes (BP), cellular components (CC) and molecular functions (MF) GO terms enriched following analyses of

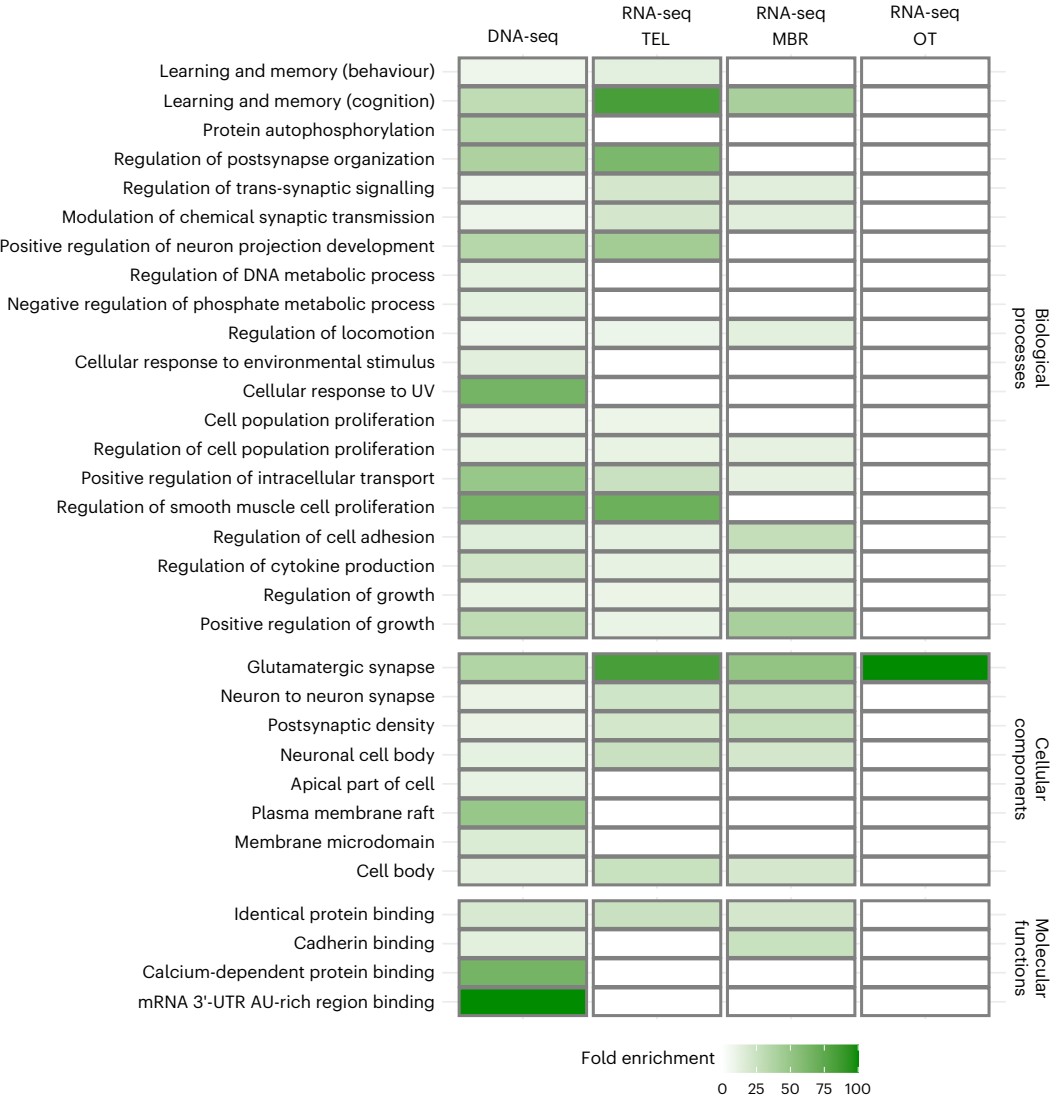

**Fig. 4 | Functional characterization of genes of interest across experiments.** Visualization of functional overlap based on GO annotations between genes of interest highlighted in strongly differentiated experimental setups evaluating social interactions of female guppies following experimental evolution for higher polarization: (1) genomic analyses of DNA comparing Pool-seq of high- and low-sociability female guppies (left column); (2) transcriptomic analyses evaluating differentially expressed genes in key brain regions of polarization-selected lines of female guppies exposed to two different social contexts: swimming alone or with a group of conspecifics (TEL, telencephalon; MBR, midbrain; OT, optic tectum; columns 2–4). Shades of green indicate fold enrichment from our statistical overrepresentation tests performed to gene lists obtained from each experiment (see Supplementary Dataset).

differentially expressed genes in the telencephalon ($n = 158$). This value represented a 1.7-fold increase in the concordance of terms in relation to mean values obtained from corresponding enrichment analyses of 1,000 random sets of 158 genes (see Methods; mean concordance (CI): 45% (43, 47)). We likewise found concordances of 64% and 4.5% for differentially expressed genes in the midbrain ($n = 109$) and in the optic tectum ($n = 21$), respectively. These represented 2.1-fold and 1.1-fold increases in relation to analyses with 1,000 random sets of 109 and 21 genes in midbrain and optic tectum, respectively (mean concordance midbrain: 30% (28, 31); mean concordance telencephalon: 3.8% (3.6, 4.2)). We summarized and visualized GO terms enrichment lists across experiments and tissues sampled using REVIGO[36]. We found a strong overlap between enrichment of GO biological process terms associated with learning and memory, synaptic processes, neuron projection and cell growth, mostly constrained to the telencephalon and midbrain regions (Fig. 4). We found similar patterns in relation to cellular component GO terms, with strong overlap in neuronal components, in particular, with high enrichment of terms associated with glutamatergic synapse. Visualization of GO terms associated with molecular functions suggests a major role of genes with protein binding function across experiments, including a role for cadherin-binding related genes in the midbrain (Fig. 4).

## Discussion

We used behavioural phenotyping across guppy families, in conjunction with Pool-seq and RNA-seq to identify the genetic architecture of coordinated motion. Our broad range of analyses, spanning genomes, transcriptomes and phenotypes, provides an exceptional evaluation of the molecular mechanisms underlying sociability in this fish. Our work suggests that genes and gene networks involved in social decision-making through neuron migration and synaptic function are key in the evolution of schooling, highlighting a crucial role of glutamatergic synaptic function and calcium-dependent signalling processes.

Our pedigree-based phenotyping analyses of 195 guppy families from polarization-selected and control lines indicate moderate levels

of heritability (Alignment: 0.06–0.34; Attraction: 0.09–0.26), with pronounced sex differences in full-pedigree models (Alignment$_{\text{female–male}}$: 0.10 ± 0.05; Attraction$_{\text{female–male}}$: −0.17 ± 0.05) in estimates for key behavioural traits forming the sociability axis in this species. Our heritability estimates are similar to estimates for affiliative social behaviour traits in primates, ungulates and rodents[10–13,37], and to overall estimates of heritability in personality traits across human and non-human animals[8,38]. Given the importance of social behaviour in a range of survival and fitness components in natural systems[1,39,40], our results suggest that complex genetic architectures can respond quickly to strong evolutionary pressures, even when only one sex is subject to selection[22], and that our lab population contained substantial amounts of standing genetic variation for these traits before selection.

The complex genetic architecture makes it difficult to precisely characterize cross-sex genetic effects in our study. We nonetheless observed a positive cross-sex genetic correlation in attraction (0.68, CI: 0.25–0.98), suggesting similarities between males and females in the genetic architecture of this trait. This result is concordant with a study focused on the bold–shy continuum aspect of personality establishing that sex differences in risk-taking behaviours are weak and probably lack sex-specific genetic architecture in this species[41]. Yet, sex differences in heritability estimates of alignment ($♀h^2_{\text{alignment}} = 0.34$ (0.18, 0.49); $♂h^2_{\text{alignment}} = 0.06$ (0.00, 0.18)) suggest that it is important to account for sex-specific additive genetic variance when inferring the evolvability of personality traits. In addition, the low cross-sex heritability we observe in these latter traits is particularly interesting and suggests that selection for a complex trait in one sex need not result in a correlated response in the other sex. Overall, this indicates significant sex-specific genetic variation for sex-specific behaviours, and that sexually dimorphic behaviours need not require decoupling of male and female genetic architecture when sufficient sex-specific genetic variation is present.

We next mapped the genomic and transcriptomic basis of phenotypic differences in polarization in female guppies. Our genome-wide association study was designed to compare individuals with high- and low-sociability phenotypes from within polarization-selected lines, rather than between polarization-selected and control lines. This may have resulted in compressed phenotypic spread but carries the important advantage of reducing the incidence of SNP frequencies that vary across alternative selection lines due to drift. As such, our design is conservative. In our most stringent Pool-seq analysis, we identified SNPs in four genes that consistently differed across all three replicate lines, these genes having been previously associated with cognition and functions relevant to social behaviour. The supt6 histone chaperone and transcription elongation factor homologue (*supt6h*) is important in the positive regulation of transcriptional elongation and a substrate of mTOR, a signalling pathway with a role in cognitive function[42,43]. The ubiquitin-specific peptidase 11 (*usp11*) homologue in humans has a critical function in the development of the neural cortex, and knockout studies in mice show that the locus protects females from cognitive impairment[44,45]. Similarly, the novel RNA-binding protein *Nova-1*-like gene is associated with a neuron-specific nuclear RNA-binding protein in humans and regulates brain-specific splicing related to synaptic function[46].

Finally, our Pool-seq analysis identified cadherin 13 (*cdh13*), the human homologue of which has a crucial role in GABAergic function[47], with involvement in neural growth and axonal guidance during early development[48,49]. Moreover, deficit of this gene has a major impact in neurodevelopmental disorders including attention-deficit/hyperactivity disorder and autism spectrum disorder[50]. Indeed, *cdh13* knockout mice display delayed acquisition of learning tasks and a decreased latency in sociability[51]. Interestingly, repeated selection of genes involved in cadherin-signalling pathways[52] has been shown in guppy populations experiencing different predation pressures. Together, natural selection imposed by differences in predation across these populations[53–55] and our combined findings in the genomic background

of guppies suggest strong selective pressures for cadherin-signalling genes due to their modulation of affiliative behaviours.

Our expression results revealed differences in regulation in genes associated with learning, behaviour and neural function, mainly in the telencephalon and midbrain, in comparisons of polarization-selected and control lines in different social contexts. Overall, this suggests that the regulation of highly demanding cognitive processes via synaptic function underlies variation in sociability. While the integration of visual signals is central in fish schools[56], our results suggest that higher-order cognitive processes are the basis of variation in social affinity. Indeed, the differences in alignment and attraction observed when swimming with unfamiliar conspecifics are arguably highly cognitively demanding, as within a collective motion context, the tendency to copy the directional movements of other individuals implies a direct trade-off between personal goal-oriented behaviours and the benefits of social conformity[57,58]. Together, our study of transcriptomic profiles of schooling fish suggests that the regulation of affiliative behaviours in this species is driven by an intricately linked social decision-making network in the brain[59], with strong links to functional groups governing social behaviours and personality across species. More broadly, our results offer insight into important questions about the evolution of behaviour and other traits with complex genetic architecture. First, our results of large-scale expression differences among selection lines are consistent with recent discussions of the role of gene regulatory networks in coordinating large numbers of genes associated with behaviours[60]. It is highly likely that the genes with convergent expression changes in the selection lines are controlled via a modular regulatory architecture, as evidenced by our co-expression network analysis (Supplementary Tables 7 and 8, and Fig. 3).

We find a striking concordance in the functionality of genes independently identified in genomic and transcriptomic profiling of strongly differentiated experiments assessing social interactions of polarization-selected female guppies (Fig. 4). The overlap in significantly enriched GO terms, including learning, synaptic processes and neuron projection restricted to brain regions associated with decision-making and motor control, strongly reinforces the notion that genetic regulation of these cognitive processes is fundamental for sociability in guppies. In addition, the functional concordance we observe between the regulatory and protein differences among our selection lines is noteworthy in the context of the discussion of whether structural or regulatory variation is more important in adaptive phenotypes[61,62]. The overlap in functionality in our genomic and transcriptomic approaches suggests that both are important, with artificial selection for behaviour acting on coding and regulatory variation within the same pathway to achieve adaptive phenotypes.

Our results indicate that the regulation of glutamatergic synaptic processes is a particularly promising network for future studies of affiliative behaviours. Interestingly, differential gene expression of glutamate receptor genes has been identified to regulate female mating preferences in guppies[63] and is concordantly identified across species of vertebrates in the regulation of long-term affiliative mating behaviours[64]. Guppies are livebearers, and this has hindered the use of functional genetic tools such as Clustered Regularly Interspaced Short Palindromic Repeats on this species. Although not feasible at this time, future functional validation via genetic manipulations of guppies for these pathways would prove extremely interesting.

Our results are also concordant with other comparative transcriptomic studies of behavioural responses towards conspecific territorial intrusions, which identified calcium ion-binding regulation across phylogenetically distant species[65]. Together, the consistency in our findings of specific genes and functional terms associated with calcium-dependent and cadherin-binding molecular functions across our experiments suggests that these are promising molecular targets for future research exploring the evolution and regulation of sociability and affiliative behaviours.

## Methods

### Ethics

All experiments were performed in accordance with ethical applications approved by the Stockholm Ethical Board (Dnr:C50/12, N173/13 and 223/15). These applications are consistent with the Institutional Animal Care and Use Committee guidelines.

### Study system

To evaluate the genetic architecture of sociability, we performed a series of experiments in guppies following artificial selection on coordinated motion. The laboratory population of guppies used originated from a downstream population of the Quare river in Trinidad, which is subject to high predation levels. The original collection was made in 1998[66] and the laboratory population has since been kept in several large (>200 l) tanks of >200 individuals each to avoid inbreeding. The artificial selection procedure is outlined in detail in refs. 22,23. In brief, groups of female guppies were subjected to repeated open field tests and were subsequently sorted on the basis of their median polarization, measured by the degree of alignment exhibited by the individuals within the group when swimming together[22,23]. For three generations, females from groups with higher polarization were mated with males from those cohorts to generate three lines of guppies that had been selected for high polarization. In parallel, random females were exposed to the same experimental conditions and were mated with unselected males to generate three control lines. Analysis of the third generation of polarization selection revealed that, on average, females exhibited a 15% higher level of polarization and a 10% higher level of group cohesiveness compared with control females[22].

Throughout the selection experiment and the completion of experiments described below, all fish were removed from their parental tanks after birth, separated by sex at the first onset of sexual maturation and afterwards kept in single-sex groups of eight individuals in 7 l tanks containing 2 cm of gravel with continuously aerated water, a biological filter and plants for environmental enrichment. We allowed for visual contact between the tanks. The laboratory was maintained at 26 °C with a 12 h light:12 h dark schedule. Fish were fed a diet of flake food and freshly hatched brine shrimp daily.

### Heritability of sociability

To investigate heritability and cross-sex genetic correlations of sociability in the guppy, we measured alignment and attraction with unfamiliar groups of conspecifics in parents and offspring from polarization-selected and control lines. Specifically, using offspring of the $F_3$ generation of selection, we bred 35 families for each of the three polarization-selected and for each of the three control lines. From our population of $F_3$ generation offspring (kept in single-sex groups before the breeding experiments), we used male and female guppies of the same age (~9 months old) and paired them in 3 l tanks to generate the parental generation. We collected offspring from the first two clutches of these pairs and transferred newborn offspring to 3 l tanks in groups of three siblings. We separated siblings by sex at the first onset of sexual maturation and afterwards kept them in single-sex groups of three individuals until behavioural testing. We phenotyped sociability for a total of 195 guppy families: mother, father and six offspring (three females and three males). Any family for which we did not collect at least three female and three male offspring was disregarded from further behavioural testing. Each of the six selection lines was represented by a minimum of 30 families in our heritability analyses.

**Behavioural assays.** To phenotype sociability in each member of our guppy families, we measured alignment and attraction of 1,495 guppies from our breeding experiment. For each fish, we performed an open field assay using white arenas with 55 cm diameter and 3 cm water depth in which our focal fish (guppies from the breeding experiment) interacted with a group of seven same-sex conspecifics. Non-focal

guppies used in these assays were from a lab wild-type stock population and of similar age to our focal fish. Before the start of the test, focal fish and the seven-fish group were acclimated in the centre of the arena for 1 min in separate opaque white 15 cm PVC cylinders. After this acclimation period, we lifted the cylinders and filmed the arena for 10 min using a Point Grey Grasshopper 3 camera (FLIR Systems; resolution, 2,048 pixels by 2,048 pixels; frame rate, 25 Hz). Three weeks before assays, we tagged wild-type fish with small black elastomere implants (Northwest Marine Technology) to allow recognition of wild-type fish after completion of each assay. After completion, we gently euthanized focal fish from the parental generation with an overdose of benzocaine and kept them in ethanol for future genomic analyses. Focal fish from the offspring generation were transferred to group tanks for future experimental use. Groups of seven wild-type fish were transferred to holding tanks and used in a maximum of seven assays with focal fish.

**Data processing.** We tracked the movement of fish groups in the collected video recordings using idTracker[67] and used fine-grained tracking data to calculate the following variables in Matlab (v.2020): (1) alignment, the median alignment of the focal fish to the group average direction across all frames in the assay. This was quantified by the total length of the sum of two-unit vectors, one representing the heading of the focal fish and the other representing the heading of the group centroid. Calculations of alignment were only obtained if six out of the eight members of the group presented tracks following the optimization of our tracking protocol in the setup in refs. 22,23,68; (2) attraction, the median nearest neighbour distance across all frames in the assay; and (3) activity; we obtained the median speed across all group members and across all frames by calculating the first derivatives of the $x$ and $y$ time series, followed by smoothing using a Savitzky–Golay filter with span of 12 frames (1/2 s) and degree 3. For all measurements, trials with less than 70% complete tracks ($n = 8$) were disregarded in further analyses. The proportion of frames used did not differ between polarization-selected and control fish for any comparison across different generations and sexes (Supplementary Fig. 5). We calculated these variables for the focal fish and the average for the seven-fish wild-type group. To recover focal fish id in the tracking data, we used idPlayer to visualize trials by projecting the raw tracking data onto experimental videos. We followed focal individuals for the first 2 min of the assay and used the stable identity assigned by idTracker in data collection. In trials with less than 85% complete tracks ($n = 8$), we followed focal individuals for the total duration of the recording to verify the consistency in identity assigned by idTracker. This approach has previously shown strong reliability in individuals that were observed using this protocol for 20 min recordings in the same experimental setup that quantified sexual behaviour of guppies in mixed-sex shoals[69].

**Statistical analyses.** Analyses were conducted using R statistical software (v.4.1.3)[70], RStudio (v.2023.3.1.446)[71] and the tidyverse package[72]. We used LMMs with alignment and attraction as dependent variables to test for potential differences between polarization-selected and control lines in social interactions with unfamiliar individuals. Selection regime, sex, the interaction between these two factors and generation were included as fixed effects. The average activity of the wild-type group was coded as a covariate, with a random intercept for each replicated selection line, the breeding family and the number of tests previously performed with the wild-type group as random factors. All models were run using lme4 and lmerTest packages[73,74]. Model diagnostics showed that residual distributions were roughly normal with no evidence of heteroscedasticity.

To estimate heritability, the degree of phenotypic variation due to genetic inheritance, and cross-sex genetic correlations of alignment and attraction, we used Bayesian animal models[75]. Animal models use a matrix of pedigree relationships set as a random effect to separate phenotypic variance for each response variable into additive genetic

variance and the remaining variance. Given strong sex differences in social interactions in guppies, we performed three animal models for each trait: one including the data on the 1,495 phenotyped individuals and two including only the phenotyped females or males. Parameter values were estimated using the brms interface[76,77] to the probabilistic programming language Stan[78]. We used normal priors with a mean of 0 and s.d. of 3 for fixed effects, and Student-$t$ priors with 5 degrees of freedom, a mean of 0 and s.d. of 5 for random effects. The full-pedigree model estimated cross-sex correlations with a Lewandowski–Kuro-wicka–Joe (LKJ) prior with $\eta = 1$, which is uniform over the range −1 to 1. Posterior distributions for full/same-sex pedigree models were obtained using Stan's no-U-turn Hamiltonian Monte Carlo with 24/16 independent Markov chains of 2,500/4,000 iterations, discarding the first 1,500/2,000 iterations per chain as warm-up and resulting in 24,000/32,000 posterior samples overall. Convergence of the chains and sufficient sampling of posterior distributions were confirmed by a potential-scale-reduction metric ($R$) below 1.01 and an effective sample size of at least 1,000. For each model, posterior samples were summarized on the basis of the Bayesian point estimate (posterior median) and posterior uncertainty intervals by Highest Density Intervals. We calculated estimates of heritability by taking the ratio of the additive genetic variance to the total phenotypic variance in each independent model (see Supplementary Tables 5 and 6).

### Genetic basis of sociability in guppies

**Pooled DNA sequencing.** We extracted DNA of muscle tissue from the caudal peduncle of polarization-selected females from the parental generation using Qiagen's DNeasy Blood and Tissue kit following standard manufacturer protocol, with an additional on-column RNase A treatment. We quantified DNA concentration using fluorometry (Qubit, ThermoFisher). We next pooled samples from the 7 females that represented the top and bottom 20% polarization-selected guppy lines whose families presented higher and lower sociability in 6 final pools at equimolar amounts (Supplementary Fig. 1). We achieved a minimum of 3 µg genomic DNA per pool. We used a Nextera DNA Flex library preparation kit (Illumina) following manufacturer protocol. The final library containing 6 pooled samples was sequenced at SciLife Lab, Uppsala (Sweden) in one lane of an Illumina NovaSeq 6000 system. We obtained on average 31.8 million 150 bp read pairs per sample (26.9 million read pairs minimum per sample).

**Read quality control and trimming.** We assessed the quality of reads for each pool using FastQC v.0.11.4 (www.bioinformatics.babraham.ac.uk/projects/fastqc). After verifying initial read quality, reads were trimmed with Trimmomatic (v.0.35)[79]. We filtered adaptor sequences and trimmed reads if the sliding window average Phred score over four bases was <15 or if the leading/trailing bases had a Phred score <4, removing reads post filtering if either read pair was <50 bases in length. Quality was verified after trimming with FastQC.

**Genome-wide allele frequency analysis.** Reads were mapped to the guppy reference genome assembly using default settings (Guppy_female_1.0 + MT, RefSeq accession: GCA_000633615.2)[80] with bwa-mem (v0.7.17)[81]. We used Samtools (v.1.6.0)[82] to convert sam to bam files, sort bam files, remove duplicates and make mpileup files. First, to identify SNPs that significantly differed in their allele frequencies between guppies with high and low sociability, we merged sequences from high-sociability and low-sociability pools and used Popoolation2 (ref. 27) to create a synchronized file with allele frequencies for high and low sociability (mpileup2sync.pl –min-qual 20), compute allele frequency differences (mpileup2sync.pl –min-count 6 –min-coverage 25 –max-coverage 200), calculate Fst for every SNP (fst-sliding.pl) and perform a Fisher's exact test (fisher-test.pl). Second, we similarly used Popoolation2 to detect consistent changes in allele frequencies of sociability pooled samples for our three replicated artificial selection lines. For this, we created one sync file per replicate (mpileup2sync.pl –min-qual 20) and performed a CMH test (cmh-test.pl –min-count 18 –min-coverage 25 –max-coverage 200). Using package qqman[83] in R (v.4.1.3)[70], we made Manhattan plots for each chromosome by plotting the negative $\log_{10}$-transformed $P$ values of the exact Fisher and CMH tests as a function of chromosome position.

**Significance tests and functional analyses.** We determined SNPs that were significantly different between high- and low-sociability merged pools in Fisher's exact tests using the traditional genome-wide significance threshold ($-\log_{10}(P) > 8$)[28]. We next used custom scripts to identify the overlap between the positions of these SNPs and genes present in the guppy reference annotated genome[80] and to find homologous genes of this set in medaka (*Oryzias latipes*). We further used this set of unique genes ($n = 160$) to determine associated GO terms between our merged pools. For this, we performed enrichment tests in PANTHER[84], as implemented in the GO Ontology Consortium (http://www.geneontology.org/). To test for enrichments of GO terms, we performed one-tail Fisher's exact tests with a Bonferroni-corrected $P$ value threshold of $P < 0.05$ using a full list of medaka genes orthologous to guppy genes as background. We used Revigo (http://revigo.irb.hr)[36] to find and visualize representative subsets of terms on the basis of semantic similarity measurements for our enriched GO terms related to biological processes, cellular components and molecular functions.

For CMH test results, we determined SNPs that were significantly different between high- and low-sociability pools based on FDR-corrected $P < 0.01$. We used a custom script to identify the overlap between the positions of these SNPs and genes present in the guppy reference annotated genome[80].

### Neurogenomic response of schooling in guppies

**Behavioural assays and tissue collection.** Using offspring of the $F_3$ generation (6 months old), we placed an individual or groups of eight unfamiliar adult control and polarization-selected females in white 55 cm arenas. After 30 min, females were euthanized by transfer to ice water. After 30 s, with the aid of a Leica S4E microscope, we removed the top of the skull and after cutting transversally posterior of the optic tectum and anterior of the cerebellum, and horizontally through the optic chiasm, removed the brain from the skull and placed it into ice water. We severed the 'telencephalon' from the rest of the brain between the ventral telencephalon and thalamus at the 'commissura anterioris', including both the pallium and subpallium regions. Then we cut the laminated cup-like structures of the 'optic tectum'. The remaining part of the brain was the 'midbrain'. Dissections took under 2 min and tissue samples were immediately preserved in RNAlater (Ambion) at 4 °C for 24 h and then at −20 °C until RNA extraction.

**RNA extraction and sequencing.** For each treatment, we pooled tissue from 10 individuals into 2 non-overlapping pools of 5 for each replicate line. We used this strategy to reduce noise in transcript expression data during sample normalization procedures, potentially caused by outliers during behavioural experiments while maintaining each replicate as a comparable unit. Our experimental design represents a total of 120 individual females, constituting 6 pools per treatment per selection regime for a total of 24 pools per tissue. Each sample pool was homogenized and RNA was extracted using Qiagen's RNeasy kits following standard manufacturer protocol. Libraries for each sample were prepared and sequenced by the Wellcome Trust Center for Human Genetics at the University of Oxford, United Kingdom. All samples were sequenced across nine lanes on an Illumina HiSeq 4000 system. We obtained on average 33.9 million 75 bp read pairs per sample (28.9 million read pairs minimum, 39.8 million maximum).

**Read quality control and trimming.** We assessed the quality of reads for each sample using FastQC v.0.11.4 (www.bioinformatics.babraham.

ac.uk/projects/fastqc). After verifying initial read quality, reads were trimmed with Trimmomatic (v.0.35)[79]. We filtered adaptor sequences and trimmed reads if the sliding window average Phred score over four bases was <15 or if the leading/trailing bases had a Phred score <3, removing reads post filtering if either read pair was <33 bases in length. Quality was verified after trimming with FastQC.

**Differential expression analysis.** We mapped RNA-seq reads against the latest release of the published guppy genome assembly[80] using the HiSat 2.0.5–Stringtie v.1.3.2 suite[81]. For each individual pool, reads were mapped to the genome and built into transcripts using default parameters. The resulting individual assemblies were then merged into a single, non-redundant assembly using the built-in StringTie-merge function. We filtered the resulting assembly for non-coding RNA using medaka and Amazon molly (*Poecilia formosa*) non-coding RNA sequences as reference in a nucleotide BLAST (Blastn) search. After eliminating all sequences matching non-coding RNAs, we kept only the longest isoform representative for each transcript for further analysis. Finally, we quantified expression by re-mapping reads to this filtered assembly using RSEM (v.1.2.20)[85].

Lowly expressed genes were removed by filtering transcripts with <2 reads per kilobase per million mapped reads, preserving only those transcripts that have expression above this threshold in at least half of the samples for each treatment within a line. After this final filter, a total of 26,140 optic tectum transcripts, 25,100 telencephalon transcripts and 26,514 midbrain transcripts were retained for further analysis. Using sample correlations in combination with multidimensional scaling plots based on all expressed transcripts, we determined that none of the 72 pools represented outliers, hence all samples were included in the analysis.

We used DESeq2 (ref. [86]) to normalize filtered read counts using standard function to identify DE genes between the Alone and the Group treatment in control and polarization-selected lines separately and then examined the overlap in differentially expressed genes between them. A transcript was considered differentially expressed if it had an FDR-corrected $P < 0.05$. As behaviour could be modulated by small changes in expression, we did not filter differentially expressed genes on the basis of log fold-change in expression between the treatments.

**Differential co-expression analysis.** We used BFDCA[30] to identify pairs of genes that have different correlation patterns in the two conditions[32,87,88]. Here we compared the Alone and Group treatments within each line for each tissue separately, in the same manner as the previously described DE analysis. BFDCA is based on weighted gene co-expression network analysis and has been shown to be a reliable and accurate method[30]. This untargeted approach to differential co-expression analysis uses a combined Bayes factor, a ratio of the marginal likelihoods of the data between the two alternative hypotheses, to evaluate which genes are differentially correlated in the two conditions. We controlled for false positives and accounted for multiple testing by integrating a random permutations approach[32]. In short, we created 1,000 permuted datasets and considered a DC gene pair significant if the Bayes factor for the actual expression data was larger than the 1% tail of the permuted data Bayes factor distribution.

**Functional analyses.** To investigate the function of DE genes, we performed GO term enrichment tests. To accomplish this, we initially completed the annotation of the reference genome assembly. The transcripts without clear gene names from the reference genome, and the de novo transcripts identified by HiSat were annotated with blastX against the Swissprot non-redundant database. We then determined which GO terms were associated with differentially expressed genes and performed BP, CC and MF enrichment tests in PANTHER[84]. To assess the level of concordance between genes of interest across experiments,

we compared the proportions of BP, CC and MF GO terms that were significantly enriched in genomic analyses of sociability implemented in polarization-selected females and the proportions of BP, CC and MF GO terms enriched in differential expression analyses in brain tissue of polarization-selected females following exposure to Group and Alone experimental conditions. To assess their significance, we compared these values to mean proportions obtained from bootstrap analyses of 1,000 random sets of 158 (for comparison with telencephalon), 109 (midbrain) and 21 (optic tectum) genes from our medaka–guppy orthologous gene list. All analyses were based on one-tail Fisher's exact tests with a Bonferroni-corrected $P$ value threshold of $P < 0.05$ using medaka genes orthologous to guppy genes as the background. Bootstrap analyses with random sets of genes were automated using rbioapi package[89] in R (v.4.1.3)[70]. We next summarized and visualized GO terms enrichment lists across experiments and tissues using REVIGO[36] (settings: SimRel semantic similarity measure, 0.5 value). To investigate the function of differentially co-expressed genes, we used g:Profiler[90] to identify the enriched BP GO terms and pathways that were altered across mating contexts associated with differentially co-expressed gene pairs. We determined overrepresented pathways among DC gene pairs in each tissue using the human (*Homo sapiens*) database in g:Profiler. We chose the human database for its completeness, acknowledging the distant phylogenetic relationship to guppies.

**Reporting summary**
Further information on research design is available in the Nature Portfolio Reporting Summary linked to this article.

## Data availability
Data needed to evaluate the conclusions in the paper are deposited in figshare under accession code https://doi.org/10.6084/m9.figshare.23805702. Genomic and transcriptomic data are deposited at NCBI under accession codes PRJNA994132 and PRJNA504011. Video recordings related to this paper may be requested from the authors. Source data are provided with this paper.

## Code availability
Codes needed to evaluate the conclusions in the paper are deposited in figshare under accession code https://doi.org/10.6084/m9.figshare.23805702.

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

## Acknowledgements

We thank D. Sumpter, K. Pelckmans and J. Herbert-Read for important contributions to the conceptualization of the artificial selection procedure; J. Shu (jacelyndesigns.com) for composing guppy graphics for figures; A. Rennie, E. Trejo and A. Boussard for help with fish husbandry. This work was supported by the Knut and Alice Wallenberg Foundation (102 2013.0072 to N.K.), the Canada 150 Research Chair Program, the European Research Council (680951 to J.E.M), the Swedish Research Council (2016-03435 to N.K., 2017-04957), the Royal Swedish Academy of Sciences (BS2019-0046 to A.C.-L.), Lars Hiertas Memorial Foundation (FO2019-0477 to A.C.-L.), European Research Council (H2020 Marie Skłodowska-Curie Actions 654699 to N.I.B) and Universidad de los Andes (FAPA-4700000443 to N.I.B).

## Author contributions

J.E.M., N.K. and A.C.-L. conceptualized and acquired funding for the project. N.K., A.K., A.S. and M.R. designed the selection procedure

and behavioural experiments. A.C.-L. conducted research to obtain behavioural data. A.C.-L. and W.v.d.B. performed formal analyses and visualization of behavioural and heritability data. A.C.-L. and M.C.-C. conducted research to obtain genomic data. A.C.-L., M.C.-C. and I.D. performed formal analyses and visualization of genomic data. N.I.B., S.D.B. and A.K. conducted research to obtain transcriptomics data. N.I.B. and. A.C.-L. performed formal analyses and visualization of transcriptomics data. A.C.-L., J.E.M. and N.K. wrote the original draft. All authors contributed to the final version of the manuscript.

## Funding

## Competing interests

The authors declare no competing interests.

## Additional information

**Correspondence and requests for materials** should be addressed to Alberto Corral-Lopez.

# Reporting Summary

## Statistics

For all statistical analyses, confirm that the following items are present in the figure legend, table legend, main text, or Methods section.

| n/a | Confirmed | |
|---|---|---|
| ☐ | ☒ | The exact sample size (*n*) for each experimental group/condition, given as a discrete number and unit of measurement |
| ☐ | ☒ | A statement on whether measurements were taken from distinct samples or whether the same sample was measured repeatedly |
| ☐ | ☒ | The statistical test(s) used AND whether they are one- or two-sided<br>*Only common tests should be described solely by name; describe more complex techniques in the Methods section.* |
| ☐ | ☒ | A description of all covariates tested |
| ☐ | ☒ | A description of any assumptions or corrections, such as tests of normality and adjustment for multiple comparisons |
| ☐ | ☒ | A full description of the statistical parameters including central tendency (e.g. means) or other basic estimates (e.g. regression coefficient) AND variation (e.g. standard deviation) or associated estimates of uncertainty (e.g. confidence intervals) |
| ☐ | ☒ | For null hypothesis testing, the test statistic (e.g. *F*, *t*, *r*) with confidence intervals, effect sizes, degrees of freedom and *P* value noted<br>*Give P values as exact values whenever suitable.* |
| ☐ | ☒ | For Bayesian analysis, information on the choice of priors and Markov chain Monte Carlo settings |
| ☐ | ☒ | For hierarchical and complex designs, identification of the appropriate level for tests and full reporting of outcomes |
| ☐ | ☒ | Estimates of effect sizes (e.g. Cohen's *d*, Pearson's *r*), indicating how they were calculated |

*Our web collection on statistics for biologists contains articles on many of the points above.*

## Software and code

Policy information about availability of computer code

| Data collection | Collection motion : idTracker (v2.1) |
|---|---|
| Data analysis | R (v4.1.3), RStudio (v2022.07.0), Matlab Compiler Runtime 8.3, Matlab (v2020a)<br>Sequence quality control: FastQC (v.0.11.4.553); Trimmomamtic (v0.35)<br>Genome-assembly: HiSat (v2.05); StringTie (v1.32)<br>Pool-Seq: Samtools (v.1.6.0); Popoolation2 (v1.1013)<br>Differential expression and Coexpression: DeSeq2 & BFCSA R packages<br>Functional analyses: PANTHER (v18.0); REVIGO (v1.8.1); gProfiler R package |

For manuscripts utilizing custom algorithms or software that are central to the research but not yet described in published literature, software must be made available to editors and reviewers. We strongly encourage code deposition in a community repository (e.g. GitHub). See the Nature Portfolio guidelines for submitting code & software for further information.

## Data

Policy information about availability of data

All manuscripts must include a data availability statement. This statement should provide the following information, where applicable:
- Accession codes, unique identifiers, or web links for publicly available datasets
- A description of any restrictions on data availability
- For clinical datasets or third party data, please ensure that the statement adheres to our policy

Data needed to evaluate the conclusions in the paper are deposited in figshare database under accession code 10.6084/m9.figshare.23805702. Genomic and transcriptomic data are deposited at NCBI database under accession codes PRJNA994132 and PRJNA504011. Source data are provided with this paper. Video recordings related to this paper may be requested from the authors.

## Research involving human participants, their data, or biological material

Policy information about studies with human participants or human data. See also policy information about sex, gender (identity/presentation), and sexual orientation and race, ethnicity and racism.

| | |
|---|---|
| Reporting on sex and gender | N/A |
| Reporting on race, ethnicity, or other socially relevant groupings | N/A |
| Population characteristics | N/A |
| Recruitment | N/A |
| Ethics oversight | N/A |

Note that full information on the approval of the study protocol must also be provided in the manuscript.

# Field-specific reporting

Please select the one below that is the best fit for your research. If you are not sure, read the appropriate sections before making your selection.

☐ Life sciences   ☐ Behavioural & social sciences   ☒ Ecological, evolutionary & environmental sciences

For a reference copy of the document with all sections, see nature.com/documents/nr-reporting-summary-flat.pdf

# Ecological, evolutionary & environmental sciences study design

All studies must disclose on these points even when the disclosure is negative.

| | |
|---|---|
| Study description | We phenotyped male and female guppies following artificial selection for higher polarization to estimate the effect on social interactions and the heritability of sociability.<br><br>We sequenced DNA pools (Pool-seq) to identify genome-wide differences in allele frequencies between polarization-selected female guppies that presented high sociability and female guppies that presented low sociability.<br><br>We use RNA sequencing to determine differences in gene expression in multiple brain regions of polarization-slelected and control females in response to two different social contexts (swimming alone or schooling in a group with seven unfamiliar females). |
| Research sample | Heritability: 1487 guppies across 195 families, including the mother, father, three female and three male offspring from replicated experimental selection lines ( three polarization-selected and three control lines)<br><br>Pool-seq: for each replicated selection line, pooled samples from seven females representing top and bottom 20% sociability scores ( 6 total pooled samples).<br><br>Transcriptomics: pooled samples of brain tissue (optic tectum, midbrain and telencephalon) of ten individuals into two non-overlapping pools of five for each replicated selection line for each treatment (alone or swimming in a group with conspecifics). |
| Sampling strategy | Heritability: Using offspring of the F3 generation of artificial selection we bred 35 families for each of replicated selection line (three polarization-selected and thre control). We used male and females of same age (9 months old approximately) and paired them in 3L tanks to generate a parental generation. We collected offspring for the two first clutches of these pairs and phenotyped three females and three males at 5-6 moths old. |

Genomics: Only samples from parental generation from heritability experiments were kept for subsequent genomic analyses. Based on sociability measurements, we used top and bottom 20% samples from each replicate (7 samples per pool - total 6 pools).

Transcriptomics: This strategy was applied to reduce noise in transcript expression data during sample normalization procedures ( because of potential outliers in behavioral experiments) and to maintain each replicate as a comparable unit.

Data collection

Heritability: A.C-L. performed experiments to record social interactions of guppies. Groups of 8 sexually mature female or male guppies were evaluated in an experimental arena using an open field test. An even number of up to 16 groups of fish were tested per day, balancing the number of polarization-selected and control groups per day. All trials were recorded and we tracked positional data of focal fish and mean values for the other 7 individuals in the test. Collective motion characteristics were extracted from tracking data.

Genomics: A.C-L. and M.C-C. extracted DNA from female samples kept in ethanol. We achieved a minimum of 3g genomic DNA per pool using Nextera DNAFlex library preparation kit (Illumina) following manufacturer protocol. The final library containing six pooled samples was sequenced at SciLife Lab, Uppsala.

Transcriptomics: A.K. and S.D.B. performed experiments and dissections for tissue collection. Fish were placed individually or in groups of 8 unfamiliar adult control and polarization-selected females in white 55cm arenas. After 30 minutes, females were euthanized by transfer to ice water and tissues of interest dissected out and kept in RNA later. N.B. extracted RNA by pooling tissue of interest, homogenizing and following Qiagen RNeasy's protocol. libraries for each sample were prepared and sequenced by the Welcome Trust for Human Genetics (Oxford University, UK).

Timing and spatial scale

Heritability: Data collection of parental generation was performed between March-April 2019 ( 8-16 trials a day - total 390 trials). We stopped data collection for 3-4 months and proceed when individuals of offspring generation sexually matured. Data collection of offspring generation was performed between September-December 2019 ( 8-16 trials a day - 1100 individuals)

Transcriptomics:  measurements and experiments were performed during Autumn 2019 ( 8 individuals per day during a three-weeks collection period).

All experiments were performed in lab facilities at Department of Zoology, Stockholm University (Sweden).

Data exclusions

Families that did not produce three female and three male offspring were disregarded from behavioral testing. We obtained data for a minimum of 30 families for each of the six selection lines (three polarization-selected, three control).

Following automated tracking, all trials  with less than 70% complete tracks (n= 8) were disregarded for further analyses.

Reproducibility

All experiments performed to polarization-selected and control lines relied on protocols successfully applied in previous investigations in this species and setup (e.g. Kotrschal, Szorkovsky et al. Sci Adv 2020). Sampling design of transcriptomic analyses was based on a successful protocol we previously used in the study of neurogenomics of mate preference in guppies (Bloch et al. Nat Eco Evo 2018). There were no attempts to repeat the experiments. Trials with incomplete data were excluded as stated above.

Randomization

Fish used in F0 of our artificial selection procedure were a mix of adult fish of various ages from the lab breeding stock. In subsequent generations we used relatively uniform young adults to minimize time between generations. This design ensured that within every replicate, polarization and control fish were of same age. Fish used for experiments in this study are offspring of similar age from selection and control lines of F3 generation.

An even number of up to 16 groups were tested per day in our experimental setups, balancing the number of polarization-selected and control groups per day

Blinding

Data acquisition was based on extraction from positional data using an automated protocol.

Did the study involve field work?    ☐ Yes    ☒ No

# Reporting for specific materials, systems and methods

We require information from authors about some types of materials, experimental systems and methods used in many studies. Here, indicate whether each material, system or method listed is relevant to your study. If you are not sure if a list item applies to your research, read the appropriate section before selecting a response.

## Materials & experimental systems

| n/a | Involved in the study |
|-----|----------------------|
| ☒ | Antibodies |
| ☒ | Eukaryotic cell lines |
| ☒ | Palaeontology and archaeology |
| ☐ ☒ | Animals and other organisms |
| ☒ | Clinical data |
| ☒ | Dual use research of concern |
| ☒ | Plants |

## Methods

| n/a | Involved in the study |
|-----|----------------------|
| ☒ | ChIP-seq |
| ☒ | Flow cytometry |
| ☒ | MRI-based neuroimaging |

# Animals and other research organisms

Policy information about studies involving animals; ARRIVE guidelines recommended for reporting animal research, and Sex and Gender in Research

| Laboratory animals | Guppies used in our experiment are laboratory-raised descendants of Trinidad guppies sampled from the high predation populations of the Quare River (Trinidad). Generation 0 of selection was a mix of adult fishes of various ages from the laboratory breeding stock, while for the next generations we used relatively uniform young adults (approximately 4-5 months-old). |
|---|---|
| Wild animals | The study did not involve wild animals |
| Reporting on sex | We used a breeding design involving male and female guppies to quantify the effect of experimental evolution of higher polarization. We report sex differences in behavioral measurements, as well as across sex correlations in heritability estimates. |
| | As, we used female guppies as the target of directional selection for higher coordination, genomic and transcriptomic studies presented here focus only on female guppies. |
| Field-collected samples | The study did not involve samples collected from the field |
| Ethics oversight | All experiments were performed in accordance with ethical applications approved by the Stockholm Ethical Board (Dnr:C50/12, N173/13, and 223/15). These applications are consistent with the Institutional Animal Care and Use Committee guidelines. |

Note that full information on the approval of the study protocol must also be provided in the manuscript.

