## [Peer Review File · Nature Ecology & Evolution]

Peer Review Information

Journal: Nature Ecology & Evolution

Manuscript Title: Functional convergence of genomic and transcriptomic architecture underlying schooling behaviour in a live-bearing fish

Corresponding author name(s): Alberto Corral-Lopez

Editorial Notes:

Reviewer Comments & Decisions:

Decision Letter, initial version:

30th May 2023

Dear Dr Corral-Lopez,

Your manuscript entitled "Functional convergence of genomic and transcriptomic genetic architecture underlying sociability in a live-bearing fish" has now been seen by 3 reviewers, whose comments are attached. The reviewers have raised a number of concerns which will need to be addressed before we can offer publication in Nature Ecology & Evolution. We will therefore need to see your responses to the criticisms raised and to some editorial concerns, along with a revised manuscript, before we can reach a final decision regarding publication.

We therefore invite you to revise your manuscript taking into account all reviewer and editor comments. Please highlight all changes in the manuscript text file.

* If you have not done so already please begin to revise your manuscript so that it conforms to our Article format instructions at <http://www.nature.com/natecolevol/info/final-submission>. Refer also to any guidelines provided in this letter.

2[REDACTED]

Nature Ecology & Evolution is committed to improving transparency in authorship. As part of our efforts in this direction, we are now requesting that all authors identified as 'corresponding author' on published papers create and link their Open Researcher and Contributor Identifier (ORCID) with their account on the Manuscript Tracking System (MTS), prior to acceptance. ORCID helps the scientific community achieve unambiguous attribution of all scholarly contributions. You can create and link your ORCID from the home page of the MTS by clicking on 'Modify my Springer Nature account'. For more information please visit www.springernature.com/orcid.

[REDACTED]

Reviewer expertise:

Reviewer #1: fish neurogenomics

Reviewer #2: genomics of social behaviour

Reviewer #3: collective behaviour, tracking

Reviewers' comments:

Reviewer #1 (Remarks to the Author):

In the submitted manuscript, the authors apply an integrative approach, coupling behavioral assays and genomic research, to identify the neurogenomic mechanisms that shape sociability in the guppy *Poecilia reticulata*. Specifically, the authors identify novel promising candidate genes across multiple brain regions implicated in regulating schooling, an affiliative behavior in fishes. Overall, I believe the content and topic is relevant for publication in Nature Ecology and Evolution. However, there are a few

2relatively minor comments worth addressing, listed below. Some of these include analyses that, although not crucial for publication, would improve the framing of the overall discussion.

SPECIFIC COMMENTS

Line 101- Should the term "sociality" be switched to "sociability" here? It's unclear if this represents a different term. If so, the distinction could be made clearer here.

Lines 113-117 - This sentence reads a bit long and could be divided into two sentences for clarity.

Line 160 – Capitalize "Methods".

Supplementary Figure S1 – There is a legend issue: Polarization should be orange, and control should be gray.

Line 203 – I believe this should read "polygenic", not "polygenetic".

Lines 202 – 204 – Is this measure of "Most significant differences..." quantifiable, or based on the spread visualized on the figure itself? If quantifiable, do you have a record of this number? This currently is unclear.

Figure 2 – This is minor since the axes themselves have no intrinsic meaning, but could you switch the presentation or labeling of these axes? Semantic space x and y are inverted in the current state.

Line 215- Tense, "looked" instead of "look".

Lines 275-285 – In its current state, this section, along with Figure 4, don't particularly highlight this level of concordance the authors are portraying. Perhaps there is a way to statistically test this by comparing the likelihood of sharing these GO terms across experiments? I think adding a formal test will strengthen the result seen in Figure 4 and described here.

Lines 324 – 341 – Although perhaps not crucial, some discussion on the gene expression patterns of genes identified in PoolSeq (cdh13, usp11, Nova-1-like, supt6h) could add to better link the PoolSeq vs RNA-Seq results. Although these genes don't appear to be differentially expressed based on Supplementary Results (I wouldn't necessarily expect them to be), it would be interesting to see if they are perhaps assigned to modules (via a network analysis like WGCNA) related to their purported functions.

Reviewer #2 (Remarks to the Author):

This is a well-designed and executed study to explore the genotype to phenotype issue, in the context

3of complex social behaviour. The study combines sophisticated behavioural sampling and analysis with well-established methods in DNA and RNA sequencing, with the key finding that the genes implicated by genomics and transcriptomics “converge” on common functional mechanisms, as inferred by Gene Ontology analysis. There is no follow-up functional genomic experimentation to test any of these inferences.

Social behaviour is a particularly challenging area for understanding the pathway from “g2p” because it's invariably a long and winding road. In that context, there is strong intrinsic and broad interest in the topic of this paper, with the expectation and hope of insights that would extend beyond the results of the paper itself. In that sense, the paper, as currently written, unfortunately does not deliver. It does not explicitly engage with insights from fish sociability that might be useful in thinking about the genotype to (behavioural) phenotype issue in a more generic sense. For a high-profile journal like NEE, I think this would be a necessary component. Knowing the great reputation and accomplishments of the senior author, I have no doubt that such insights could be derived from this paper and developed.

To continue in this vein, why is it noteworthy that some of the same genes with SNPS also show expression variation? There are other cases when this is not seen. Is there any rhyme or reason that is starting to develop in the field for whether DNA variation needs to manifest as difference in expression or not?

Lesser issues

The study is based on selected lines; in recounting the history of those lines (from previously published work), it is clear that there is a lot of standing variation for the behavioural traits in question. This reader, at least, would be interested in thoughts on why that is. (As it relates to previously published work, this request might be beyond the scope of the present ms)

Some of the experimental conditions associated with the behavioural work were not clear. Age of animals; nature of the environmental conditions and how they relate to natural conditions (including predator environment).

DNA results showed only 13 out of 421 showing consistency among replicates of the same selected lines. This reviewer did not find any commentary on this finding. On one hand focusing on the 13 provides a “highly filtered” set to base the subsequent analyses. On the other hand, given that these are replicates of the same selected line, what might be the sources of this variation or inconsistency (depending on one's perspective)?

As noted above, there are no functional genomics experiments to confirm the “convergence” findings. This is always a difficult issue for polygenic traits like behaviour. This reviewer thinks that this manuscript should at least address the issue in thoughtful discussion.

Methods: The text says that it was necessary to pool individuals to obtain sufficient RNA for RNAseq. Given that it is not possible to routinely do single cell RNAseq, this comment is confusing. And especially problematic given the sacrifice of individual variation, which always is important for behaviour.

4Reviewer #3 (Remarks to the Author):

Dear Editor and authors,

Thank you for giving me the opportunity to review this paper. The study focuses on using guppies as a biological model to investigate the heritability of sociality across sexes, examining interindividual variation and evolutionary processes. The research is conducted systematically and presents intriguing results. However, I do have a few comments regarding the definition of "sociality" in collectives and the specifics of the data collection process.

Major comments:

1. In this paper, the authors have chosen alignment and attraction as the primary variables to assess sociality in collectives. I am curious about the rationale behind this choice, as relying solely on average alignment and attraction values may not necessarily capture the full essence of "sociality." The introduction mentions the affiliation phenomenon, indicating that exploring the data from additional perspectives, such as examining the frequency and duration of affiliative interactions, could provide valuable insights. Furthermore, the mathematical definition of "higher polarization" lacks clarity. It would be helpful to understand how the authors calculate polarization and what specific threshold is employed to determine higher polarization. Additionally, in the calculation of polarization, it would be worthwhile to consider if any filters are applied, such as including only connected groups with an interindividual distance of less than 10 cm and excluding tracking data shorter than 16 consecutive tracked frames, similar to the analyses conducted on offspring data.

2. In relation to tracking and identification methods, the authors mentioned tagging for non-focal individuals, but it is unclear whether tagging was also implemented for the offspring. If tagging was not used for offspring identification, how did the authors differentiate and identify individual offspring? Relying solely on Idtracker may not provide reliable results, and it would be valuable to know the reliability of the tracking method employed. Additionally, including an example of the tracked data would help support the justification of the tracking approach.

Minor comments:

1. On line 153, it would be beneficial to provide data rather than citing a paper to support the claim that differences in body size between age classes in guppies may explain the observed results.
2. The method section would benefit from including detailed information on how higher polarization was determined. Please elaborate on the specific methods used for calculating higher polarization.
3. On line 422, it would be helpful to specify the duration of the acclimation period in the experimental procedure.
4. On line 438, it is necessary to provide a justification for using a 10 cm threshold for defining a connected group. Additionally, it would be insightful to discuss the potential impact and sensitivity of the results to changes in this parameter.
5. On line 439, it would be valuable to provide information on the frequency of missing data for at least 16 consecutive tracked frames. Furthermore, it would be important to explain the approach

5taken when the missing data is less than 16 consecutive tracked frames. Is interpolation employed in such cases? Additionally, a rationale for selecting 16 consecutive tracked frames instead of 25 frames (equivalent to 1 second) should be provided.

*****END*****

Author Rebuttal to Initial comments

We have endeavoured to address all comments, and we provide point-by-point responses below to all reviewer comments (in bold) with edits implemented in the manuscript (colored in blue in this response letter). We thank the reviewers for their insightful and constructive suggestions, which have significantly improved the manuscript.

Reviewer expertise:

Reviewer #1: fish neurogenomics

Reviewer #2: genomics of social behaviour

Reviewer #3: collective behaviour, tracking

Reviewers' comments:

Reviewer #1 (Remarks to the Author):

In the submitted manuscript, the authors apply an integrative approach, coupling behavioral assays and genomic research, to identify the neurogenomic mechanisms that shape sociability in the guppy *Poecilia reticulata*. Specifically, the authors identify novel promising candidate genes across multiple brain regions implicated in regulating schooling, an affiliative behavior in fishes. Overall, I believe the content and topic is relevant for publication in Nature Ecology and Evolution. However, there are a few relatively minor comments worth addressing, listed below. Some of these include analyses that, although not crucial for publication, would improve the framing of the overall discussion.

6We thank the reviewer for the kind words about our work. We have addressed all comments and analyses suggested by the reviewer and we hope that editor and reviewers agree with us that they have contributed to strengthen the quality of the manuscript.

SPECIFIC COMMENTS

Line 101- Should the term “sociality” be switched to “sociability” here? It’s unclear if this represents a different term. If so, the distinction could be made clearer here.

Yes, we have changed to sociability in the text.

Lines 113-117 - This sentence reads a bit long and could be divided into two sentences for clarity.

We have changed accordingly (Lines 113-118):

‘Understanding the genetic basis of this schooling phenotype requires linking individual phenotypic differences to genetic variation. In this study, we phenotyped alignment and attraction of 1496 guppies across 195 families (father, mother, three female and three male offspring from our replicate experimental selection lines) to estimate the heritability of these two motion characteristics that previous factor analyses identified integral components for the sociability axis of personality in this species²¹.’

Line 160 – Capitalize “Methods”.

We have changed it.

Supplementary Figure S1 – There is a legend issue: Polarization should be orange, and control should be gray.

We have edited accordingly

Line 203 – I believe this should read “polygenic”, not “polygenetic”.

We have changed it.

Lines 202 – 204 – Is this measure of “Most significant differences...” quantifiable, or based on the spread visualized on the figure itself? If quantifiable, do you have a record of this number? This currently is unclear.

We agree that the phrasing in the previous version of our manuscript might have led to confusion on how we delineated significance in our GWAS analyses, and we apologize for this. We defined SNPs as significantly different if their p-value was lower than the significance threshold commonly used in GWAS studies, after accounting for multiple comparisons across the genome ($p < 10^{-8}$). We have rephrased this accordingly (Lines 202-205):

‘Using this methodology, we identified 819 SNPs associated with our sociability phenotype (Fisher’s exact test, $p < 10^{-8}$; Fig 2A). SNPs over this standard genome-wide significance threshold²⁸ were mostly found in single physically unlinked positions across the genome, consistent with a polygenic architecture of the trait.’

Figure 2 – This is minor since the axes themselves have no intrinsic meaning, but could you switch the presentation or labeling of these axes? Semantic space x and y are inverted in the current state.

We agree and have changed accordingly in figure 2 and Supplementary Figures.

Line 215- Tense, “looked” instead of “look”.

We have changed it.

Lines 275-285 – In its current state, this section, along with Figure 4, don’t particularly highlight this level of concordance the authors are portraying. Perhaps there is a way to statistically test this by comparing the likelihood of sharing these GO terms across experiments? I think adding a formal test will strengthen the result seen in Figure 4 and described here.

Many thanks for this excellent suggestion! We now provide a formal test of the level of convergence between enriched GO terms found across experiments. For this, we have evaluated the proportion of GO terms that are enriched both in our experiments with DNA-Poolseq and in our differentially expressed genes in RNA-seq analyses. Next, we compared these values to those obtained from bootstrap analyses implemented using 1000 random sets of similar number of genes as those found differentially expressed for each comparison. Our results now combine the use of summary and visualization tools with these analyses showcasing larger fold-increases in the proportion of genes simultaneously enriched across experiments in relation than those obtained with random sets of genes. We have added the following information in relation to this in the Methods and Results section:

1- Results (Lines 295-316):

'Specifically, we used functional analyses in the set of genes with differentiated SNPs between merged sequences of the three replicates with high and low sociability (273 unique genes) as reference, and compared it to functional analyses of genes differentially expressed in three different brain tissues of females following exposure to multiple social conditions. We found a concordance of 79% in the combination of biological processes (BP), cellular components (CC) and molecular functions (MF) GO terms enriched following analyses of differentially expressed genes in the telencephalon (n =158). This value represented a 1.7-fold increase in the concordance of terms in relation to mean values obtained from corresponding enrichment analyses of 1000 random sets of 158 genes (see Methods; mean concordance \pm [CIs]: 45% [43,47]). We likewise found concordances of 64% for differentially expressed genes in the midbrain (n = 109), and of 4.5% for differentially expressed genes in the optic tectum (n = 21). These represented 2.1-fold and 1.1-fold increases in relation to analyses with 1000 random sets of 109 and 21 genes in midbrain and optic tectum respectively (mean concordance midbrain: 30% [28,31]; mean concordance telencephalon: 3.8% [3.6,4.2]). We summarized and visualized GO terms enrichment lists across experiments and tissues sampled using REVIGO³⁵. We found a strong overlap between enrichment of GO biological process terms associated with learning and memory, synaptic processes, neuron projection and cell growth, mostly constrained to the telencephalon and midbrain regions (Fig. 4). We found similar patterns in relation to cellular component GO terms, with strong overlap in neuronal components, in particular with high enrichment of terms associated with glutamatergic synapse. Visualization of GO terms associated with molecular functions suggests a major role of genes with protein binding function across experiments, including a role of cadherin-binding related genes in the midbrain (Fig. 4).'

2- Methods (Lines 673-691):

'Functional Analyses. To investigate the function of DE genes we performed GO term enrichment tests. To accomplish this, we initially completed the annotation of the reference genome assembly. The transcripts without clear gene names from the reference genome, and the de novo transcripts identified by HiSat were annotated with blastX against the Swissprot non-redundant database. We then determined which GO terms were associated with differentially expressed genes and performed Biological Processes (BP), Cellular Components (CC) and Molecular Functions (MF) enrichment tests in PANTHER⁸³. To assess the level of concordance between genes of interest across experiments, we compared the proportion of BP, CC and MF GO terms that were significantly enriched in genomic analyses of sociability implemented in polarization-selected females, and the proportion of BP, CC and

9MF GO terms enriched in differential expression analyses in brain tissue of polarization-selected females following exposure to Group and Alone experimental conditions. To assess their significance, we compared these values to mean proportions obtained from bootstrap analyses of 1000 random sets of 158 (for comparison with telencephalon), 109 (midbrain) and 21 (optic tectum) genes from our medaka-guppy orthologous gene list. All analyses were based on one-tail Fisher's exact tests with a Bonferroni corrected p-value threshold of $p < 0.05$ using medaka genes orthologous to guppy genes as the background. Bootstrap analyses with random sets of genes were automated using rbioapi package⁸⁸ in R (v4.1.3)⁶⁹. We next summarized and visualized GO terms enrichment lists across experiments and tissues using REVIGO³⁵ (settings: SimRel semantic similarity measure, 0.5 value).'

Lines 324 – 341 – Although perhaps not crucial, some discussion on the gene expression patterns of genes identified in PoolSeq (cdh13, usp11, Nova-1-like, supt6h) could add to better link the PoolSeq vs RNA-Seq results. Although these genes don't appear to be differentially expressed based on Supplementary Results (I wouldn't necessarily expect them to be), it would be interesting to see if they are perhaps assigned to modules (via a network analysis like WGCNA) related to their purported functions.

We have now performed differential coexpression analyses with RNA-seq data. Genes identified by Pool-seq analyses were not present in differentially coexpressed gene pairs across our analyses. However, we believe that the findings from these new analyses strengthen our major point of convergence in the function of genes highlighted from results obtained in Pool-Seq and differential expression analyses. Specifically, that genes involved in glutamatergic synaptic function and protein binding signaling (especially those related to calcium-dependent processes) play an important role in modulating sociability in this species. As such, we have now incorporated the following changes to the manuscript:

- 1- We added major findings from these analyses to results and discussion and provide extended details of genes highlighted in these analyses in Supplementary materials (Lines 268-288):

‘Differential coexpression analyses. We used systems biology methods designed to compare the coexpression networks between conditions to identify genes that change in the way they are connected to other genes within the coexpression network across conditions, independent of whether they are differentially expressed^{28–30}. Specifically, we used BFDCA²⁹ to identify differential coexpressed (DC) gene pairs under the Group and Alone conditions (i.e. pairs of genes that significantly change in correlation between the two social contexts for each line^{29,31}). Similar to findings in DE analyses, we found little overlap in the genes forming DC gene pairs between comparisons implemented for control and polarization-selected lines (see Supplementary Tables 7-8, Fig. 3). Together, our results suggest that polarization-selected lines were activating different biological pathways than control lines to modulate coordinated movement.

We additionally found a group of genes that are both differentially expressed (DE) and differentially coexpressed (DC) in the same tissue and line, suggesting they might play an important role mediating coordinated movement (Supplementary Tables 8-9). Specifically, in the telencephalon, we identified 4 genes that are both DE and DC in polarization-selected lines: *LRRC24*, *PTPRS*, *KHDR2*, and *PP2BA* (Supplementary Table 9). These genes are part of the Calcineurin and the Wnt/Oxytocin signalling pathways, known to be involved in modulating social behavior, learning and memory^{32–34}. Enrichment tests confirm the functional relevance of the DC gene pairs identified, revealing an overrepresentation of genes associated with the glutamatergic synapse, as well as visual transduction among DC gene pairs in multiple comparisons (Supplementary Tables 10-11).’

- 2- We have edited figure 3 incorporating the overlap between differentially expressed genes (DE) and differentially coexpressed (DC) gene pairs, as well as the overlap between DC gene pairs in comparisons for polarization-selected and control lines.

Figure 3. Neurogenomic response of schooling in guppies. Hierarchical clustering and relative expression level for all differentially expressed genes between Alone and Group treatments in the (a) optic tectum, (b) the telencephalon and (c) the midbrain. Differentially expressed genes were identified separately in polarization and control line samples. Clustering, based on Euclidian distance, represents transcriptional similarity across all samples, with bootstrap values (1000 replicates) shown at nodes. Venn diagrams summarize the total number of differentially expressed (DE) genes and differentially coexpressed (DC) gene pairs in each tissue for polarization-selected and control lines.

3- We have incorporated edits to describe the methods used to perform these analyses (Lines 6661-671 and 691-696):

'Differential Coexpression Analysis. We used Bayes approach for Differential Coexpression Analysis (BFDCA)²⁹, in order to identify pairs of genes that have different correlation patterns in two conditions^{31,86,87}. Here we compared the Alone and Group treatments within each line for each tissue

separately, in the same manner as the previously described DE analysis. BFDCA is based on WGCNA and has been shown to be a reliable and accurate method²⁹. This untargeted approach to differential coexpression (DC) analysis uses a combined Bayes factor, a ratio of marginal likelihood of the data between the two alternative hypotheses, to evaluate which genes are differentially correlated in two conditions. We controlled for false positives and accounted for multiple testing by integrating a random permutations approach³¹. In short, we created 1000 permuted datasets and considered a DC gene pair significant if the Bayes factor for the actual expression data was larger than the 1% tail of the permuted data Bayes factor distribution. ...

... To investigate the function of differentially coexpressed genes, we used g:Profiler⁸⁹ to identify the enriched BP GO terms and pathways that were altered across mating contexts associated with differentially coexpressed gene pairs. We determined over-represented pathways among DC gene pairs in each tissue using the human (*Homo sapiens*) database in g:Profiler. We chose the human database for its completeness, acknowledging the distant phylogenetic relationship to guppies.’

Reviewer #2 (Remarks to the Author):

This is a well-designed and executed study to explore the genotype to phenotype issue, in the context of complex social behaviour. The study combines sophisticated behavioural sampling and analysis with well-established methods in DNA and RNA sequencing, with the key finding that the genes implicated by genomics and transcriptomics “converge” on common functional mechanisms, as inferred by Gene Ontology analysis. There is no follow-up functional genomic experimentation to test any of these inferences.

Many thanks for the kind remarks. We agree that follow up functional experiments would be a major advance. Unfortunately, at this time, gene editing in guppies is simply not possible, as their internal gestation complicates stable genetic modification through systems such as CRISPR, although advances may make this possible at some point in the future. More importantly, as the reviewer notes below, functional genetic validation of complex behaviours underpinned by polygenic architecture is notoriously fraught, as it would require modification of dozens, if not hundreds of genes. We would need to the study the effects of these myriad of genes both on their own as well as part of a network, requiring many years of study. As such, much as we agree that it would be fascinating, we respectfully suggest that it is just not possible to comply with this reviewer suggestion at this time.

13Social behaviour is a particularly challenging area for understanding the pathway from "g2p" because it's invariably a long and winding road. In that context, there is strong intrinsic and broad interest in the topic of this paper, with the expectation and hope of insights that would extend beyond the results of the paper itself. In that sense, the paper, as currently written, unfortunately does not deliver. It does not explicitly engage with insights from fish sociability that might be useful in thinking about the genotype to (behavioural) phenotype issue in a more generic sense. For a high-profile journal like NEE, I think this would be a necessary component. Knowing the great reputation and accomplishments of the senior author, I have no doubt that such insights could be derived from this paper and developed.

The senior author thanks the reviewer for the kind remarks. We have endeavoured to build up the genome-phenotype discussion for social behaviour in a way that does not bog down the discussion. In addition to the revisions associated with the comment below, we believe that there are two main broader insights that our manuscript presents. The first is to do with gene regulatory networks and behaviour (Lines 394-401):

‘More broadly, our results offer insight into important questions about the evolution of behaviour, and other traits with complex genetic architecture. First, our results of large-scale expression differences among selection lines are consistent with recent discussions of the role of gene regulatory networks in co-ordinating large numbers of genes associated with behaviours⁵⁹. It is highly likely that the genes with convergent expression changes in the selection lines are controlled via a modular regulatory architecture, as evidenced by our coexpression network analysis (Supplementary Tables 7-8, Fig. 3).’

The second comes from the cross-sex genetic correlations (Lines 348-353):

‘Additionally, the low cross-sex heritability we observe in these latter traits is particularly interesting, and suggests that selection in one sex for a complex trait need not result in a correlated response in the other sex. Overall, this indicates significant sex-specific genetic variation for sex-specific behaviours, and that sexually dimorphic behaviors need not require decoupling of male and female genetic architecture when sufficient sex-specific genetic variation is present.’

Admittedly, we are unsure whether these insights are what the reviewer was hoping for, and would welcome any further detailed suggestions.

To continue in this vein, why is it noteworthy that some of the same genes with SNPs also show expression variation? There are other cases when this is not seen. Is there any rhyme or reason that is starting to develop in the field for whether DNA variation needs to manifest as difference in expression or not?

Thanks for this suggestion. The key importance of this result is in the context of the longstanding debate about whether structural (e.g. protein sequence) or regulatory variation is more important in adaptive evolution. The convergence of our DNA and RNA datasets suggests that both are important, with artificial selection acting on both coding and regulatory variation within the same pathway, to achieve adaptive phenotypes.

To that end, we have added the following to the discussion (Lines 408-414):

‘Additionally, the functional concordance we observe between the regulatory and protein differences among our selection lines is noteworthy in the context of the discussion of whether structural or regulatory variation is more important in adaptive phenotypes^{60,61}. The overlap in functionality in our genomic and transcriptomic approaches suggests that both are important, with artificial selection for behavior acting on coding and regulatory variation within the same pathway to achieve adaptive phenotypes.’

Lesser issues

The study is based on selected lines; in recounting the history of those lines (from previously published work), it is clear that there is a lot of standing variation for the behavioural traits in question. This reader, at least, would be interested in thoughts on why that is. (As it relates to previously published work, this request might be beyond the scope of the present ms)

This is an interesting point, although one that in theory applies to selection experiments in any organism. In the response to the comment below, we mention that the lab population has been kept in several large tanks, to avoid inbreeding and associated reductions in genetic diversity. Beyond that rather standard procedure, we are unsure whether the guppies are in any way unusual compared to other animal models used in artificial selection in their level of standing variation. We have added the following to the discussion (Lines 334-338):

‘Given the importance of social behavior in a range of survival and fitness components in natural systems^{1,38,39}, our results suggest that complex genetic architectures can respond quickly to strong evolutionary pressures, even when only one sex is subject to selection²², and that our lab population contained significant amounts of standing genetic variation for these traits prior to selection.’

Some of the experimental conditions associated with the behavioural work were not clear. Age of

animals; nature of the environmental conditions and how they relate to natural conditions (including predator environment).

This information is contained in references provided for detailed work on the artificial selection procedure. However, we agree that is a good idea to incorporate this information here as well in addition to our brief description of the procedure and we have added the following (Lines 435-439 and 451-457):

‘To evaluate the genetic architecture of sociability, we performed a series of experiments in guppies following artificial selection on coordinated motion. The laboratory population of guppies used originated from a downstream population of the Quare river in Trinidad, which is subject to high predation levels. The original collection was made in 1998⁶⁵, and the laboratory population has since been kept in several large (>200-litre) tanks of >200 individuals each to avoid inbreeding. The artificial selection procedure is outlined in detail in^{22,23}. ...

... Throughout the selection experiment and the completion of experiments described below all fish were removed from their parental tanks after birth, separated by sex at the first onset of sexual maturation, and afterwards kept in single-sex groups of eight individuals in 7-L tanks containing 2 cm of gravel with continuously aerated water, a biological filter, and plants for environmental enrichment. We allowed for visual contact between the tanks. The laboratory was maintained at 26°C with a 12-h light:12-h dark schedule. Fish were fed a diet of flake food and freshly hatched brine shrimp daily.’

DNA results showed only 13 out of 421 showing consistency among replicates of the same selected lines. This reviewer did not find any commentary on this finding. On one hand focusing on the 13 provides a “highly filtered” set to base the subsequent analyses. On the other hand, given that these are replicates of the same selected line, what might be the sources of this variation or inconsistency (depending on one’s perspective)?

This is an interesting point. We do expect substantial levels of genetic drift in each selection line separately. By comparing the replicate lines and identifying the convergent changes (e.g. 13 sites), it is possible to filter out sites changing through drift alone. We have added the following to the results (Line 217):

‘Convergent changes in allele frequency likely represent selected sites, and are less likely the result of genetic drift in any one line.’

As noted above, there are no functional genomics experiments to confirm the “convergence” findings. This is always a difficult issue for polygenic traits like behaviour. This reviewer thinks that this manuscript should at least address the issue in thoughtful discussion.

We have added the following to the discussion (lines 420-422):

‘Guppies are livebearers, and this has hindered the use of functional genetic tools such as CRISPR on this species. Although not feasible at this time, future functional validation via genetic manipulations of guppies of these pathways would prove extremely interesting.’

Methods: The text says that it was necessary to pool individuals to obtain sufficient RNA for RNAseq. Given that it is not possible to routinely do single cell RNAseq, this comment is confusing. And especially problematic given the sacrifice of individual variation, which always is important for behaviour.

We thank the reviewer for this comment. In fact, the pooling strategy we employ helps us reduce individual variation, and any results are therefore more likely to be statistically robust when comparing selection lines. This strategy of course means that we cannot compare individual variation. However, the highly polygenic nature and subtle expression differences of the selection lines suggest that individual comparisons of behaviour and expression would be extremely noisy, and unlikely to yield robust results. In our previous experience with studies of guppy brain gene expression (e.g. Chen et al. 2015; Bloch et al. 2018), similar pooling strategies were highly effective at reducing individual transcriptional noise to reveal important biological signal. We have clarified our motivations behind our experimental design in the manuscript (Lines 615-620):

‘For each treatment, we pooled tissue from ten individuals into two non-overlapping pools of five for each replicate line. We used this strategy to reduce noise in transcript expression data during sample normalization procedures potentially caused by outliers during behavioral experiments, while maintaining each replicate as a comparable unit. Our experimental design represents a total of 120 individual females, constituting six pools per treatment, per selection regime for a total of 24 pools per tissue.’

References cited in the response

Bloch NI, Corral Lopez A, Buechel SD, Kotrschal A, Kolm N, Mank JE (2018) Early neurogenomic response associated with variation in guppy female preference. *Nature Ecology & Evolution* 2: 1772-1781

Chen Y-C, **Harrison PW**, Kotrschal A, Kolm N, **Mank JE**, Panula P (2015) Expression change in *Angiopoietin-1* underlies change in relative brain size in fish. *Proceedings of the Royal Society, B*. 282: 20150872

Reviewer #3 (Remarks to the Author):

Dear Editor and authors,

Thank you for giving me the opportunity to review this paper. The study focuses on using guppies as a biological model to investigate the heritability of sociality across sexes, examining interindividual variation and evolutionary processes. The research is conducted systematically and presents intriguing results. However, I do have a few comments regarding the definition of "sociality" in collectives and the specifics of the data collection process.

We appreciate the kind words and thank the reviewer for the work and suggestions.

1. In this paper, the authors have chosen alignment and attraction as the primary variables to assess sociality in collectives. I am curious about the rationale behind this choice, as relying solely on average alignment and attraction values may not necessarily capture the full essence of "sociality." The introduction mentions the affiliation phenomenon, indicating that exploring the data from additional perspectives, such as examining the frequency and duration of affiliative interactions, could provide valuable insights.

We apologize for any lack of clarity in our previous version, as our intention was not to use alignment and attraction in this study to assess sociality. Our previous version of the manuscript only referred to sociality once and this has been corrected now following comments from reviewer 1. We instead use alignment and attraction to assess the sociability axis of this species based on findings presented in Sumpter et al. (2018). Specifically, factor analyses over multiple observational and empirical data in guppies identified that these two measurements are the two major components of the sociability axis in the species and that the study of sociability provides a useful framework to identify collective behavior patterns in this and other species. We have clarified this in the Introduction (lines 113-118):

'Understanding the genetic basis of this schooling phenotype requires linking individual phenotypic variation to genetic differences. In this study, we phenotyped alignment and attraction of 1496 guppies across 195 families (father, mother, three female and three male offspring from our replicate experimental selection lines) to estimate the heritability of these two motion characteristics that

18previous factor analyses identified to be integral components for the sociability axis of personality in this species²¹.

Reference cited in the response

Sumpter, D. J., Szorkovszky, A., Kotrschal, A., Kolm, N., & Herbert-Read, J. E. (2018). Using activity and sociability to characterize collective motion. *Philosophical Transactions of the Royal Society B: Biological Sciences*, 373(1746), 20170015.

Furthermore, the mathematical definition of "higher polarization" lacks clarity. It would be helpful to understand how the authors calculate polarization and what specific threshold is employed to determine higher polarization. Additionally, in the calculation of polarization, it would be worthwhile to consider if any filters are applied, such as including only connected groups with an interindividual distance of less than 10 cm and excluding tracking data shorter than 16 consecutive tracked frames, similar to the analyses conducted on offspring data.

We think our previous description of the methods might have led to the misinterpretation that different sets of data were collected from parents and offspring and that we compared polarization of different generations in our study. However, in this study we did not quantify polarization, this is the collective motion characteristic that we used to generate the selection lines. In this study, we focused on motion characteristics at the individual level and quantified alignment to group direction and attraction (nearest neighbor distance) of parents and offspring when swimming with a group of same-sex conspecifics.

We now briefly describe in the manuscript with more clarity how polarization was quantified in previous work and how we bred individuals based on this group measurement, as well as providing references to extended methods for these. As specified above, the major focus of the study is the assessment of alignment and attraction at the individual level, including measurements of heritability in this trait. Our methods describe now in detail how we quantified these traits in 195 males and 195 females from the F3 generation of selection of our polarization-selected and control lines, as well as the quantification of these measurements using same methodology in 6 offspring resulting from pairs formed with these 195 males and 195 females. We hope that the edits have now clarified this important difference in the manuscript. We have likewise revised the methods for data collection including the application of data filters for the collected data (Lines 475-479 and 492-504):

‘To phenotype sociability in each member of our guppy families, we measured alignment and attraction of 1495 guppies from our breeding experiment. For each fish, we performed an open field assay using white arenas with 55 cm diameter and 3 cm water depth in which our focal fish (guppies from the breeding experiment) interacted with a group of seven same-sex conspecifics...

... **Data processing.** We tracked the movement of fish groups in the collected video recordings using idTracker⁶³ and used fine-grained tracking data to calculate the following variables in Matlab (v2020): i) alignment, the median alignment of the focal fish to the group average direction across all frames in the assay. This was quantified by the total length of the sum of two-unit vectors, one representing the heading of the focal fish, and one representing the heading of the group centroid. Calculations of alignment were only obtained if six out of the eight members of the group presented tracks following the optimization of our tracking protocol in the setup in (22,23,64); ii) attraction, the median nearest neighbor distance across all frames in the assay; and iii) activity, we obtained the median speed across all group members across all frames by calculating the first derivatives of the x and y time series, then smoothed using a third-order Savitzky–Golay filter. For all measurements, trials with less than 70% complete tracks (n = 8) were disregarded for further analyses. The proportion of frames used did not differ between polarization-selected and control fish for any comparison across different generations and sexes (Supplementary Figure 5).’

2. In relation to tracking and identification methods, the authors mentioned tagging for non-focal individuals, but it is unclear whether tagging was also implemented for the offspring. If tagging was not used for offspring identification, how did the authors differentiate and identify individual offspring? Relying solely on Idtracker may not provide reliable results, and it would be valuable to know the reliability of the tracking method employed. Additionally, including an example of the tracked data would help support the justification of the tracking approach.

We apologize for the lack of clarity with the use of tagging. We use these tags to accurately recover polarization-selected and control fish from the test for subsequent genomic analyses (parents) or future experiments (offspring). Tagging was implemented in wild-type fish and had no relationship with tracking data collection. Our previous experience using idTracker in our setup and following individuals for behavioral observations suggested that individual identity provided by the software is very accurate for recordings with high reliability (over 85% of frames tracked). We have clarified this in the manuscript adding information on the reliability of this method based on previous experience with it in the setup for a different experiment (Lines 479-489 and 500-510):

“Prior to the start of the test, focal fish and the seven-fish group were acclimated in the centre of the arena for one minute in separate opaque white 15 cm PVC cylinders. After this acclimation period, we lifted the cylinders and filmed the arena for 10 minutes using a Point Grey Grasshopper 3 camera (FLIR Systems; resolution, 2048 pixels by 2048 pixels; frame rate, 25 Hz). Three weeks prior to assays, we tagged wild-type fish with small black elastomere implants (Northwest Marine Technology Inc.) to allow recognition of wild-type fish after completion of each assay. After completion, we gently euthanized

20focal fish from the parental generation with an overdose of benzocaine and kept them in ethanol for future genomic analyses. Focal fish from the offspring generation were transferred to group tanks for future experimental use. Groups of seven wild-type fish were transferred to holding tanks and used in a maximum of seven assays with focal fish...

... For all measurements, trials with less than 70% complete tracks ($n = 8$) were disregarded for further analyses. The proportion of frames used did not differ between polarization-selected and control fish for any comparison across different generations and sexes (Supplementary Figure 5). We calculated these variables for the focal fish and the average for the seven-fish wild-type group. To recover focal fish id in the tracking data we used idPlayer to visualize trials by projecting the raw tracking data onto experimental videos. We followed focal individuals for the first two minutes of the assay and used the stable identity assigned by idTracker in data collection. In trials with less than 85% complete tracks ($n = 8$), we followed focal individuals for the total duration of the recording to verify the consistency of identity assigned by idTracker. This approach has previously shown strong reliability in individuals that were observed using this protocol for 20-min recordings in the same experimental setup that quantified sexual behavior of guppies in mixed-sex shoals⁶⁹

We have additionally provided data on the reliability of tracking data across treatments including a supplementary figure with this information:

Figure S5. Proportion of frames obtained from idTracker software in tracking from videos recorded for groups of seven wild-type guppies and one individual from either polarization-selected or control selection lines. Tracking data was used next for the quantification of alignment, attraction and speed in parents and six offspring of 195 families. We found no significant differences for the proportion of frames used for videos used for polarization-selected and control lines in a Linear Mixed Model that included generation, sex and selection line as fixed factors (LMM_{frames}: line: $F = 0.82$; $df = 1464$; $p = 0.37$). Horizontal lines indicate medians, boxes indicate the interquartile range, and whiskers indicate all points within 1.5 times the interquartile range. P-values in top position of each comparison indicate values for statistical contrasts of the model by sex and generation. Consistency in identity assigned by idTracker was verified in trials with less than 85% frames tracked (dashed line).'

Minor comments:

1. On line 153, it would be beneficial to provide data rather than citing a paper to support the claim that differences in body size between age classes in guppies may explain the observed results.

We agree, we now provide data on body size differences across age classes in our data set in a supplementary table:

Table S3. Mean body size of guppies used to estimate the heritability of alignment and attraction at the time of testing across sexes and generations. Values are provided in number of pixels obtained from idTracker data using a custom script implemented in Matlab (v2020a) that extracts body size data accounting for changes in apparent size between the middle and edge of the experimental arena used for video recordings.'

Generation	Selection line	Sex	Mean body size	SE
Parents	Control	Females	410	47
		Males	192	29
	Polarization-selected	Females	416	71
		Males	189	37
Offspring	Control	Females	247	55
		Males	158	33
	Polarization-selected	Females	251	60
		Males	160	26

2. The method section would benefit from including detailed information on how higher polarization was determined. Please elaborate on the specific methods used for calculating higher polarization.

We hope our revisions to the major points above and associated edits to the manuscript have now clarified that polarization was not quantified in this study.

3. On line 422, it would be helpful to specify the duration of the acclimation period in the experimental procedure.

This information is now clarified in the manuscript (Line 480-481):

‘Prior to the start of the test, focal fish and the seven-fish group were acclimated in the centre of the arena for one minute in separate opaque white 15 cm PVC cylinders.’

4. On line 438, it is necessary to provide a justification for using a 10 cm threshold for defining a connected group. Additionally, it would be insightful to discuss the potential impact and sensitivity of the results to changes in this parameter.

We apologize for the inaccurate description of data extraction in our previous manuscript. For the measurements in this manuscript, we did not define connected groups using a 10 cm threshold. This threshold was used to calculate group position (based on the largest subgroup) in previous studies evaluating collective motion in the setup. This threshold did not affect the calculation of alignment to group direction in the current manuscript. Instead, calculations of alignment were only obtained if six out of the eight members of the group presented tracks following the optimization of the tracking protocol in the setup in previous studies. As presented in response to a previous comment, we now provide information on the frequency of missing frames for different treatments. As such, we think it is unlikely that any systematic bias is affecting differences in alignment observed between these groups. We have now revised the manuscript accordingly (Lines 492-498):

‘We tracked the movement of fish groups in the collected video recordings using idTracker⁶⁶ and used fine-grained tracking data to calculate the following variables in Matlab (v2020): i) alignment, the median alignment of the focal fish to the group average direction across all frames in the assay. This was quantified by the total length of the sum of two-unit vectors, one representing the heading of the focal fish, and one representing the heading of the group centroid. Calculations of alignment were only obtained if six out of the eight members of the group presented tracks following the optimization of our tracking protocol in the setup in (22,23,68);’

5. On line 439, it would be valuable to provide information on the frequency of missing data for at least 16 consecutive tracked frames. Furthermore, it would be important to explain the approach taken when the missing data is less than 16 consecutive tracked frames. Is interpolation employed in such cases? Additionally, a rationale for selecting 16 consecutive tracked frames instead of 25 frames (equivalent to 1 second) should be provided.

We are thankful to the reviewer for these last two comments that has allowed us to implement important edits in our previous description of the tracking methods. We reiterate our apology for the misleading description that we filtered data based on the need to have 16 consecutive frames for a track. This mistake was caused by a miscommunication between authors of how the data extraction

24was implemented and about the application of a Savitzky-Golay filter that it was implemented during data extraction from tracking. This was only implemented to smooth median speed data across series and not for measurements of alignment or attraction. We implemented this filter in an alternative study performed in this setup where data was extracted using R software instead of Matlab. For data extraction in R, the Savitzky-Golay filtering procedure needed data from 13 consecutive frames to work (not 16 as wrongly stated). This requirement does not apply to data extraction with Matlab performed in this study and we have revised accordingly (Lines 499-501):

...‘and iii) activity, we obtained the median speed across all group members and across all frames by calculating the first derivatives of the x and y time series, then smoothed using a third-order Savitzky–Golay filter.’

Decision Letter, first revision:

22nd August 2023

Dear Dr. Corral-Lopez,

Thank you for submitting your revised manuscript "Functional convergence of genomic and transcriptomic genetic architecture underlying sociability in a live-bearing fish" (NATECOLEVOL-23040767A). It has now been seen again by the original reviewers and their comments are below. The reviewers find that the paper has improved in revision, and therefore we'll be happy in principle to offer publication in Nature Ecology & Evolution, pending some minor revisions to satisfy the reviewers' final requests and to comply with our editorial and formatting guidelines.

In particular, please note the final comments by Reviewer 3 and incorporate their suggestions in a revised version of your manuscript. Although these are minor comments, I would request you to prepare another point-by-point response as this will help us ensure that the reviewer's requests have been met.

Please email us a copy of the manuscript file in an editable format (Microsoft Word or LaTeX), as we can not proceed with PDFs at this stage. You can incorporate Reviewer 3's suggestions in this file.

We are now performing detailed checks on your paper and will send you a checklist detailing our editorial and formatting requirements in about a week. Please do not upload the final materials to the submission system or make any further revisions (apart from those requested by Reviewer 3) until you receive this additional information from us.

25[REDACTED]

Reviewer #1 (Remarks to the Author):

Thank you for revising the manuscript. I believe that the added analyses based on reviewer comments have improved the overall suitability of this article for publication in Nature Ecology & Evolution. I have no further concerns or comments for improvement to provide at this point.

Reviewer #2 (Remarks to the Author):

The revised manuscript has addressed my concerns.

Reviewer #3 (Remarks to the Author):

Thanks for the revisions. This version offers greater clarity in terms of data collection and analysis. I only have a few minor comments:

1. Even though "higher polarization" is a comparison between the treatment and control groups, it would still be beneficial to report the actual value.
2. Please specify the window size used in the Savitzky-Golay filtering.
3. The unit for body size in Table S3 appears to be missing."

Our ref: NATECOLEVOL-23040767A

13th September 2023

Dear Dr. Corral-Lopez,

Thank you for your patience as we've prepared the guidelines for final submission of your Nature Ecology & Evolution manuscript, "Functional convergence of genomic and transcriptomic genetic architecture underlying sociability in a live-bearing fish" (NATECOLEVOL-23040767A). Please carefully follow the step-by-step instructions provided in the attached file, and add a response in each row of the table to indicate the changes that you have made. Please also check and comment on any additional marked-up edits we have proposed within the text. Ensuring that each point is addressed will help to ensure that your revised manuscript can be swiftly handed over to our production team.

26****We would like to start working on your revised paper, with all of the requested files and forms, as soon as possible (preferably within two weeks). Please get in contact with us immediately if you anticipate it taking more than two weeks to submit these revised files.****

In recognition of the time and expertise our reviewers provide to Nature Ecology & Evolution's editorial process, we would like to formally acknowledge their contribution to the external peer review of your manuscript entitled "Functional convergence of genomic and transcriptomic genetic architecture underlying sociability in a live-bearing fish". For those reviewers who give their assent, we will be publishing their names alongside the published article.

Nature Ecology & Evolution offers a Transparent Peer Review option for new original research manuscripts submitted after December 1st, 2019. As part of this initiative, we encourage our authors to support increased transparency into the peer review process by agreeing to have the reviewer comments, author rebuttal letters, and editorial decision letters published as a Supplementary item. When you submit your final files please clearly state in your cover letter whether or not you would like to participate in this initiative. Please note that failure to state your preference will result in delays in accepting your manuscript for publication.

Cover suggestions

We welcome submissions of artwork for consideration for our cover. For more information, please see our https://www.nature.com/documents/Nature_covers_author_guide.pdf target="new"> guide for cover artwork.

Nature Ecology & Evolution has now transitioned to a unified Rights Collection system which will allow our Author Services team to quickly and easily collect the rights and permissions required to publish your work. Approximately 10 days after your paper is formally accepted, you will receive an email in providing you with a link to complete the grant of rights. If your paper is eligible for Open Access, our Author Services team will also be in touch regarding any additional information that may be required

27to arrange payment for your article.

Please note that *Nature Ecology & Evolution* is a Transformative Journal (TJ). Authors may publish their research with us through the traditional subscription access route or make their paper immediately open access through payment of an article-processing charge (APC). Authors will not be required to make a final decision about access to their article until it has been accepted. [Find out more about Transformative Journals](https://www.springernature.com/gp/open-research/transformative-journals)

Authors may need to take specific actions to achieve [compliance with funder and institutional open access mandates](https://www.springernature.com/gp/open-research/funding/policy-compliance-faq). If your research is supported by a funder that requires immediate open access (e.g. according to [Plan S principles](https://www.springernature.com/gp/open-research/plan-s-compliance)) then you should select the gold OA route, and we will direct you to the compliant route where possible. For authors selecting the subscription publication route, the journal's standard licensing terms will need to be accepted, including [self-archiving-and-license-to-publish](https://www.nature.com/nature-portfolio/editorial-policies/self-archiving-and-license-to-publish). Those licensing terms will supersede any other terms that the author or any third party may assert apply to any version of the manuscript.

[REDACTED]

[REDACTED]

Reviewer #1:

Remarks to the Author:

Thank you for revising the manuscript. I believe that the added analyses based on reviewer comments have improved the overall suitability of this article for publication in *Nature Ecology & Evolution*. I have no further concerns or comments for improvement to provide at this point.

Reviewer #2:

28Remarks to the Author:

The revised manuscript has addressed my concerns.

Reviewer #3:

Remarks to the Author:

Thanks for the revisions. This version offers greater clarity in terms of data collection and analysis. I only have a few minor comments:

1. Even though "higher polarization" is a comparison between the treatment and control groups, it would still be beneficial to report the actual value.
2. Please specify the window size used in the Savitzky-Golay filtering.
3. The unit for body size in Table S3 appears to be missing."

Author Rebuttal, first revision:

Reviewer #1 (Remarks to the Author):

Thank you for revising the manuscript. I believe that the added analyses based on reviewer comments have improved the overall suitability of this article for publication in Nature Ecology & Evolution. I have no further concerns or comments for improvement to provide at this point.

Reviewer #2 (Remarks to the Author):

The revised manuscript has addressed my concerns.

Reviewer #3 (Remarks to the Author):

Thanks for the revisions. This version offers greater clarity in terms of data collection and analysis. I only have a few minor comments:

1. Even though "higher polarization" is a comparison between the treatment and control groups, it would still be beneficial to report the actual value.
2. Please specify the window size used in the Savitzky-Golay filtering.
3. The unit for body size in Table S3 appears to be missing."

We are glad that reviewers think this revised version has improved in clarity and thank them for their contribution to it.

Following comments from Reviewer 3, we have implemented the following edits in the manuscript:

- We have added this information in instances that did not report directly in the text (e.g Results section – Line 134 below, and figure caption 1):

“For this, we assessed sociability in 740 females and 746 males from multiple families of three replicate lines artificially selected for a 15% average higher polarization (polarization-selected lines hereon) and three replicate control lines exposed to a group of non-kin unfamiliar conspecifics in an open field test.”

- We now provide extended details in the Savitzky-Golay filtering method (Line 501):

“we obtained the median speed across all group members and across all frames by calculating the first derivatives of the x and y time series, then smoothed using a Savitzky–Golay filter with span of 12 frames (1/2 second) and degree 3.

- We have added the body size unit to Supplementary Table 3

Final Decision Letter:

12th October 2023

Dear Alberto,

We are pleased to inform you that your Article entitled "Functional convergence of genomic and transcriptomic architecture underlies schooling behaviour in a live-bearing fish", has now been accepted for publication in Nature Ecology & Evolution.

Over the next few weeks, your paper will be copyedited to ensure that it conforms to Nature Ecology and Evolution style. Once your paper is typeset, you will receive an email with a link to choose the

30appropriate publishing options for your paper and our Author Services team will be in touch regarding any additional information that may be required

Due to the importance of these deadlines, we ask you please us know now whether you will be difficult to contact over the next month. If this is the case, we ask you provide us with the contact information (email, phone and fax) of someone who will be able to check the proofs on your behalf, and who will be available to address any last-minute problems . Once your paper has been scheduled for online publication, the Nature press office will be in touch to confirm the details.

Acceptance of your manuscript is conditional on all authors' agreement with our publication policies (see www.nature.com/authors/policies/index.html). In particular your manuscript must not be published elsewhere and there must be no announcement of the work to any media outlet until the publication date (the day on which it is uploaded onto our web site).

Please note that *Nature Ecology & Evolution* is a Transformative Journal (TJ). Authors may publish their research with us through the traditional subscription access route or make their paper immediately open access through payment of an article-processing charge (APC). Authors will not be required to make a final decision about access to their article until it has been accepted. [Find out more about Transformative Journals](https://www.springernature.com/gp/open-research/transformative-journals)

Authors may need to take specific actions to achieve [compliance](https://www.springernature.com/gp/open-research/funding/policy-compliance-faqs) with funder and institutional open access mandates. If your research is supported by a funder that requires immediate open access (e.g. according to [Plan S principles](https://www.springernature.com/gp/open-research/plan-s-compliance)) then you should select the gold OA route, and we will direct you to the compliant route where possible. For authors selecting the subscription publication route, the journal's standard licensing terms will need to be accepted, including [self-archiving-and-license-to-publish](https://www.nature.com/nature-portfolio/editorial-policies/self-archiving-and-license-to-publish). Those licensing terms will supersede any other terms that the author or any third party may assert apply to any version of the manuscript.

An online order form for reprints of your paper is available at <https://www.nature.com/reprints/author->

reprints.html"><https://www.nature.com/reprints/author-reprints.html>. All co-authors, authors' institutions and authors' funding agencies can order reprints using the form appropriate to their geographical region.

We welcome the submission of potential cover material (including a short caption of around 40 words) related to your manuscript; suggestions should be sent to Nature Ecology & Evolution as electronic files (the image should be 300 dpi at 210 x 297 mm in either TIFF or JPEG format). Please note that such pictures should be selected more for their aesthetic appeal than for their scientific content, and that colour images work better than black and white or grayscale images. Please do not try to design a cover with the Nature Ecology & Evolution logo etc., and please do not submit composites of images related to your work. I am sure you will understand that we cannot make any promise as to whether any of your suggestions might be selected for the cover of the journal.

You can generate the link yourself when you receive your article DOI by entering it here: http://authors.springernature.com/share.

[REDACTED]

P.S. Click on the following link if you would like to recommend Nature Ecology & Evolution to your librarian <http://www.nature.com/subscriptions/recommend.html#forms>

** Visit the Springer Nature Editorial and Publishing website at www.springernature.com/editorial-and-publishing-jobs for more information about our career opportunities. If you have any questions please click here.**